# MELK is an oncogenic kinase essential for mitotic progression in basal-like breast cancer cells

Yubao Wang[1,2], Young-Mi Lee[1†], Lukas Baitsch[1,2], Alan Huang[3], Yi Xiang[1,2], Haoxuan Tong[1,2], Ana Lako[1,2], Thanh Von[1,2], Christine Choi[1,4], Elgene Lim[5], Junxia Min[3], Li Li[3], Frank Stegmeier[3], Robert Schlegel[3], Michael J Eck[1,2], Nathanael S Gray[1,2], Timothy J Mitchison[6], Jean J Zhao[1,2]*

[1]Department of Cancer Biology, Dana-Farber Cancer Institute, Boston, United States; [2]Department of Biological Chemistry and Molecular Pharmacology, Harvard Medical School, Boston, United States; [3]Novartis Institutes for Biomedical Research, Cambridge, United States; [4]Harvard University, Cambridge, United States; [5]Department of Medical Oncology, Dana-Farber Cancer Institute, Boston, United States; [6]Department of Systems Biology, Harvard Medical School, Boston, United States

**Abstract** Despite marked advances in breast cancer therapy, basal-like breast cancer (BBC), an aggressive subtype of breast cancer usually lacking estrogen and progesterone receptors, remains difficult to treat. In this study, we report the identification of MELK as a novel oncogenic kinase from an in vivo tumorigenesis screen using a kinome-wide open reading frames (ORFs) library. Analysis of clinical data reveals a high level of MELK overexpression in BBC, a feature that is largely dependent on FoxM1, a master mitotic transcription factor that is also found to be highly overexpressed in BBC. Ablation of MELK selectively impairs proliferation of basal-like, but not luminal breast cancer cells both in vitro and in vivo. Mechanistically, depletion of MELK in BBC cells induces caspase-dependent cell death, preceded by defective mitosis. Finally, we find that Melk is not required for mouse development and physiology. Together, these data indicate that MELK is a normally non-essential kinase, but is critical for BBC and thus represents a promising selective therapeutic target for the most aggressive subtype of breast cancer.

*For correspondence: jean_zhao@dfci.harvard.edu

Present address: †Division of Pharmacology, Hanmi Innovative Research Center, Hanmi Pharmaceutical Company, Gyeonggi-do, Republic of Korea.

## Introduction

Breast cancer is a heterogeneous disease with a high degree of diversity in histology, therapeutic response, and treatment outcomes. Transcriptional profiling analyses have reproducibly identified at least five major 'intrinsic' subtypes of breast cancer: normal breast-like, luminal A, luminal B, HER2/Neu-enriched, and basal-like breast cancer (BBC) (*Perou et al., 2000*; *Sorlie et al., 2001*). These molecular subtypes have recently been confirmed in a comprehensive characterization of human breast tumors at the genomic, epigenetic, transcriptomic, and proteomic levels (*Cancer Genome Atlas Network, 2012*). Among these subtypes, basal-like breast cancer (BBC) is strongly associated with an aggressive phenotype and poor prognosis (*Rakha et al., 2008*). Unlike their luminal counterparts, BBC cells lack expression of estrogen receptor (ER) and progesterone receptor (PR). Most BBC tumors also lack expression of HER2 and thus this subtype largely overlaps with the clinically defined 'triple-negative' breast cancer (TNBC), which is also characterized by the lack of ER, PR, and HER2 expression (*Rakha et al., 2008*; *Foulkes et al., 2010*). The lack of these molecular targets renders BBC or TNBC cells relatively unresponsive to the targeted therapies that are highly effective in the treatment of

**eLife digest** Not all cancers are the same. There are, for example, at least five types of breast cancer. Different types of cancer can have different mutations and express different genes that determine how aggressively the tumors grow and how well they respond to different therapies. By exploiting these differences, scientists have developed therapies that target specific tumor types, and these targeted therapies have proven useful against most breast cancers.

One type of breast cancer, however, has proven hard to treat. Basal-like breast cancer grows rapidly and there are few treatment options for women with this type of cancer. One reason for this is that, unlike other forms of breast cancer, these cancers do not have the hormone receptors that are the targets of existing therapies.

Enzymes called kinases are promising alternate targets, and many kinase-inhibiting drugs can kill tumor cells in mice. Nevertheless, it has proven difficult to develop kinase inhibitors that are safe for use in humans because these drugs can also kill normal cells. To avoid this side effect, cancer researchers have been searching for a kinase that is active in cancer cells but not in normal cells.

Wang et al. tested a large collection of kinases and found that one called MELK caused tumors to grow in the mammary glands of mice. Further examination of tumor samples collected from hundreds of women in previous clinical studies revealed that MELK expression was increased in basal-like breast cancers and other breast cancer tumors that lack the usual hormone receptor targets.

When Wang et al. treated tumor cells and mice with tumors with a chemical that stops MELK working, basal-like breast cancer cells stopped multiplying and died. On the other hand, tumor cells that had the usual hormone receptors continued to multiply. To see if MELK is important in healthy mice, Wang et al. genetically engineered mice to delete the MELK gene and found that these mutant mice appear normal. The next challenge will be to test if drugs that inhibit MELK can kill basal-like breast cancer cells without having the side effect of harming normal cells.

luminal or HER2 positive breast cancer. Thus, establishing the molecular pathogenesis of this subtype and identifying potential targets for treatment remains a key challenge for BBC/TNBC.

Kinases comprise a large family of proteins that is frequently involved in tumor pathogenesis. Indeed, a large number of mutations, alterations in copy number, and/or expression level have been observed in genes encoding kinases across multiple types of human cancers. In addition, kinases have proven to be pharmacologically tractable, making inhibition of kinase activity with small molecules a highly effective strategy for cancer treatment (*Zhang et al., 2009*). Therefore, identifying kinases critical for the growth and survival of BBC cells could not only provide valuable insights into the pathogenesis of BBC, but also define potential druggable targets for therapeutic interventions.

Kinases that regulate progression through mitosis, including Aurora A, Aurora B and PLK1, are essential for cell proliferation. Inhibiting them in cancer cells causes mitotic arrest and/or abnormalities in chromosome segregation and cytokinesis, which in turn trigger apoptosis (*Taylor and Peters, 2008*; *Lens et al., 2010*). Inhibitors of these kinases are effective at eradicating human cancer cells in culture and in mouse xenograft models, but their efficacy in the clinic has been limited by killing of normal proliferating cells especially the bone marrow (*Dar et al., 2010*). If a kinase exists that is required for mitosis in a specific type of cancer cell, but not other tumor cells or in normal cells, inhibitors of that kinase might make highly effective and safe drugs. To date, this type of cancer-specific mitotic kinase has not been identified for any cancer.

In this study, we report the identification of MELK as a novel oncogenic kinase that emerged from an in vivo tumorigenesis screen. Analyses of breast cancer patient data according to subtypes revealed a remarkable overexpression of MELK in BBC. We further demonstrate that MELK is directly regulated by the FoxM1 transcription factor, a master mitotic regulator also found to be overexpressed in BBC. We discover that MELK is essential in basal-like, but not in luminal breast cancer cells. Notably, mice in which MELK has been genetically ablated display normal development and hematopoiesis. Together, our data establish MELK as a mitosis-regulating kinase involved in the pathogenesis of BBC and a promising molecular target for patients with basal-like breast malignancy.

## Results

### Identification of oncogenic kinases that contribute to tumorigenesis of HMECs in vivo

Transformation of primary human cells with defined genetic elements is a powerful method for identifying specific genes or pathways that are involved in oncogenic transformation (*Hahn et al., 1999*; *Zhao et al., 2004*). To this end, we first developed an in vivo tumorigenesis system that models the pathogenesis of human breast cancer, using a previously established human mammary epithelial cell (HMEC)-based transformation system (*Zhao et al., 2003*). To further optimize this system, we engineered telomerase-immortalized HMECs to express a dominant negative form of p53 (p53DD), NeuT and PI3KCA H1047R. The resulting cells, termed HMEC-DD-NeuT-PI3KCA, were fully transformed as evaluated by their ability to form orthotopic tumors in the mammary fat pads of mice (*Figure 1—figure supplement 1*). Our model recapitulates the concurrent activation of HER2/Neu and PI3KCA that is prevalent in breast cancer (*Stephens et al., 2012*).

The HMEC transformation system described above provided us with a platform to identify novel oncogenic events capable of replacing the mutant PIK3CA in cooperating with NeuT to drive HMECs to form tumors in mice. To this end, we infected HMEC-DD-NeuT cells (lacking the mutant PIK3CA) with subpools of a kinome-wide retroviral library consisting of 354 human kinases and kinase-related open reading frames (*Boehm et al., 2007*). The library was screened as a series of subpools of 10–12 kinase ORFs in HMEC-DD-NeuT cells. The infected cells were injected into the inguinal mammary fat pads of mice, and recipient mice were followed for tumor formation. Kinases in 12 pools induced tumor formation with latencies of 2–4 months. Genomic DNA was extracted from harvested tumor specimen as well as HMECs infected with matched pools of kinases prior to injection. We then used quantitative PCR to determine the relative abundance of each kinase in these paired samples. In total, 26 kinases were found specifically enriched in the tumors in vivo (*Figure 1*, *Figure 1—figure supplement 2*). Several candidate kinases that scored in the screen have previously been implicated as proto-oncogenes or cancer-associated genes, such as the inhibitor of nuclear factor kappa-B kinase subunit epsilon (IKBKE) (*Boehm et al., 2007*), rearranged during transfection (RET) (*Takahashi et al., 1985*), casein kinase 1 epsilon (CSNK1E) (*Kim et al., 2010*), NIMA-related serine/threonine kinase 6 (NEK6) (*Nassirpour et al., 2010*), and polo-like kinase 1 (PLK1) (*Liu et al., 2006*). At least three of them, PLK1 (*Golsteyn et al., 1994*), NEK6 (*Yin et al., 2003*), and MELK (maternal embryonic leucine zipper *kinase)* (*Le Page et al., 2011*), have been previously implicated in regulating mitotic progression.

### MELK is highly overexpressed in human breast cancer and its overexpression strongly correlates with poor disease outcomes

One of the top-scoring hits from our genetic screen was *MELK* (*Figure 1*), an atypical member of AMPK serine/threonine kinase family (*Lizcano et al., 2004*). While little is known about the exact biological functions of MELK, this kinase has been reported to be overexpressed in a variety of tumors (*Gray et al., 2005*). When we analyzed MELK expression in the breast cancer data set of The Cancer Genome Atlas (TCGA) (*Cancer Genome Atlas Network, 2012*), a large cohort consisting of 392 invasive ductal breast carcinomas and 61 samples of normal breast tissues, the level of MELK transcript was approximately eightfold higher in breast tumors compared to their normal counterparts (*Figure 2A*). The p value for this differential expression ($4.6 \times 10^{-54}$) places MELK in the top 1% overexpressed genes in breast cancer (*Figure 2A*). The overexpression of MELK in breast tumors relative to normal breast tissues was further confirmed by analyzing two other independent data sets (*Figure 2—figure supplement 1A*; *Ma et al., 2009*; *Richardson et al., 2006*).

To gain insights into the potential relevance of MELK overexpression in breast cancer, we asked whether MELK expression correlates with the status of disease. By analyzing gene expression data across five independent cohorts totaling more than 1500 patients (*Desmedt et al., 2007*; *Hatzis et al., 2011*; *Schmidt et al., 2008*; *Wang et al., 2005b*; *Supplementary file 1*), we found that higher expression of MELK was strongly associated with higher histologic grade in breast cancer (*Figure 2B*, *Figure 2—figure supplement 1B*); the p values for this correlation rank in the top 1% of a total 12,624 or more genes measured in all these cohorts.

We also examined whether MELK expression is correlated with metastatic recurrence. We analyzed three independent cohorts in which patients with early-stage breast cancer were followed for metastasis-free survival and had not received adjuvant systemic treatment after surgery (*van 't Veer et al., 2002;*

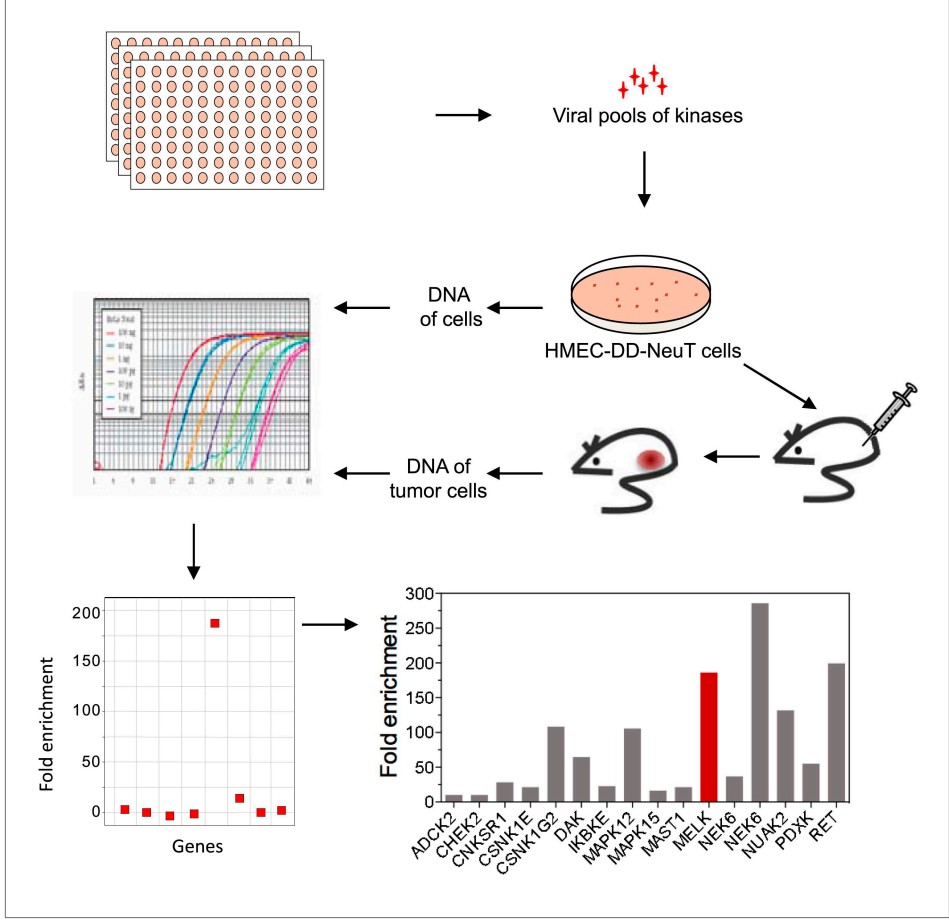

**Figure 1**. An in vivo kinome-wide screen identifies MELK as a potential oncogenic kinase. Pools of retroviral vectors encoding 354 human kinases and kinase-related proteins (37 pools in total, each consisting of 10–12 unique open reading frames) were transduced into HMED-DD-NeuT cells. After selection with neomycin, cells were transplanted into mammary fat pads of nude mice. Tumors that formed from HMECs infected with 12 pools of kinases were harvested, and genomic DNA was extracted. qPCR was performed on genomic DNA from the tumor specimens and cells infected with matched pools of kinases before injection. The relative fold enrichment was calculated from the differences in Ct value.

The following figure supplements are available for figure 1:

**Figure supplement 1**. Development of an in vivo tumorigenesis model.

**Figure supplement 2**. Screen hits and their gene description.

*Wang et al., 2005b*; *Schmidt et al., 2008*; *Supplementary file 1*). In all three cohorts, higher MELK expression levels were strongly associated with earlier metastasis in women initially diagnosed with lymph-node-negative tumors (all p values<0.001, hazard ratios >2; *Figure 2C*, *Figure 2—figure supplement 1C*). We further analyzed two cohorts, where a majority of patients had high grade and lymph-node-positive breast cancer and nearly all patients received neoadjuvant chemotherapy and/or hormone therapy (*Hatzis et al., 2011*; *Loi et al., 2007*; *Supplementary file 1*). Again, high expression level of MELK robustly correlates with metastasis in breast cancer patients (both p values<0.001, hazard ratios >2; *Figure 2C*). Thus MELK overexpression appears to have a strong predictive value for breast cancer metastasis irrespective of tumor grade or treatment regimen.

We next asked if MELK expression also correlates with the survival of breast cancer patients. In five independent large cohorts in which more than 1100 total patients were followed for overall survival (*Desmedt et al., 2007*; *Esserman et al., 2012*; *Kao et al., 2011*; *Pawitan et al., 2005*; *van de Vijver*

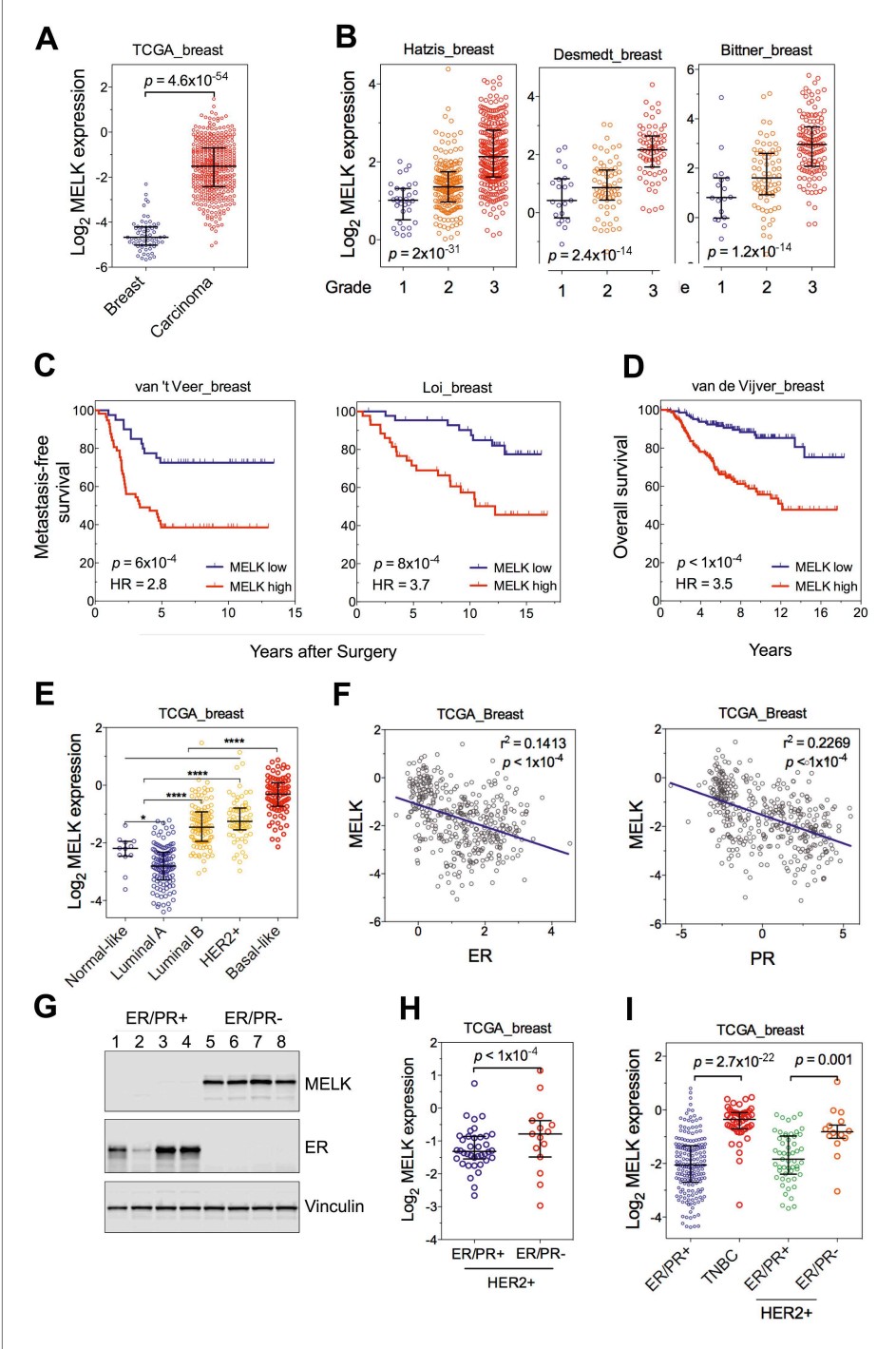

**Figure 2**. MELK is highly overexpressed in breast cancer and its overexpression strongly correlates with poor prognosis. (**A**) MELK expression levels are significantly higher in breast carcinoma (n = 392, red circles) than in normal breast tissues (n = 61, blue circles) in the TCGA breast cancer cohort (*Cancer Genome Atlas Network, 2012*). Black lines in each group indicate median with interquartile range. p=4.6 × 10$^{-54}$ (Student's *t* test). (**B**) Expression level of MELK tightly correlates with the pathological grade of breast tumors in the three independent cohorts for which these data are available. Black lines in each group indicate median with interquartile range. p values were calculated with one-way ANOVA. (**C**) Kaplan–Meier analysis of metastasis-free survival of breast cancer patients in two independent cohorts. Samples were divided into two groups with high and low expression levels of MELK. p values were obtained from the log-rank test. Hazard ratio (HR) was calculated using GraphPad Prism.

*Figure 2. Continued on next page*

*Figure 2. Continued*

(**D**) Kaplan–Meier analysis of overall survival in the van de Vijver cohort breast cancer patients. Samples were divided as in (**C**). Log-rank p value and hazard ratio (HR) are shown. (**E**) MELK expression among the molecular subtypes of breast cancer. Samples in each cohort were classified into five distinct molecular subtypes using PAM50 (*Parker et al., 2009*). Black lines in each group indicate median with interquartile range. (**F**) MELK expression inversely correlates with that of estrogen receptor (ER) or progesterone receptor (PR). Linear regression was determined using GraphPad Prism. The linear regression Pearson's correlation coefficient ($R^2$) and its p value are indicated. (**G**) ER/PR− breast tumors have higher abundance of MELK protein than ER/PR+ ones. Lysates of primary human tumors were subjected to immunoblotting using the indicated antibodies. (**H**) Expression of ER/PR determines MELK expression within HER2+ breast cancer. Samples with molecular HER2+ status were classified into ER/PR+ and ER/PR− groups. Black lines in each group indicate median with interquartile range. (**I**) MELK expression in subtypes of breast cancer that are defined by ER/PR, and HER2 expression. Note that HER2+ tumors were divided into ER/PR+ and ER/PR− groups. *p<0.05, ****p<0.0001 (Student's *t* test).

The following figure supplements are available for figure 2:

**Figure supplement 1**. MELK is a top-ranking overexpressed gene in breast cancer and a strong prognostic indicator.

**Figure supplement 2**. Correlation of MELK expression with breast cancer subtypes and the histologic grade of disease.

*et al., 2002*; *Supplementary file 1*), high expression level of MELK strongly correlated with increased rates of mortality (all p values<0.05, hazard ratios >2) (*Figure 2D*, *Figure 2—figure supplement 1D*). Together, these data show that MELK may serve as a prognostic indicator in predicting breast cancer patients' likelihood of metastasis and overall survival rate.

## MELK is commonly overexpressed in the subtype of basal-like breast tumors

Given the heterogeneity of breast cancer, we analyzed MELK expression in different subtypes of breast cancer as defined by gene expression profiling (*Perou et al., 2000*; *Sorlie et al., 2001*). We categorized samples in multiple breast cancer data sets by PAM50 gene signature (*Parker et al., 2009*). In five independent cohorts with more than 1500 patients in total, we observed a strikingly similar pattern of MELK expression among these different subtypes of breast tumors (*Figure 2E*, *Figure 2—figure supplement 1E*). While luminal A and normal-like subtypes displayed the lowest expression of MELK, basal-like breast cancers (BBC) showed the highest expression level of MELK among all subtypes (p<0.0001). Given that there are more high-grade tumors in the BBC than the other subtypes, we sought to determine the correlation of MELK with subtypes of breast tumors within the same grade. We performed statistical analysis of a large cohort of breast cancer for grade 1, 2 and 3 across all subtypes, respectively, and found that MELK is most highly expressed in BBC (*Figure 2—figure supplement 2A*), suggesting that MELK expression is most pronounced in basal-like breast tumors with the same pathological grade. Moreover, a significant association of MELK expression with disease status also exists within the subtype of BBC (*Figure 2—figure supplement 2B*), suggesting that MELK expression is associated with tumor aggressiveness and poor prognosis in this disease.

Consistent with this observation of MELK overexpression in BBC, we found that MELK expression in breast tumors has a significant inverse correlation with the expression of luminal markers, including estrogen and progesterone receptors (ER, PR) (*Figure 2F*, *Figure 2—figure supplement 1F*). To confirm this observation at the protein level, we analyzed primary tumors samples for MELK expression. Strikingly, all the four ER/PR+ tumor samples lacked detectable signal of MELK expression. In contrast, ER/PR-negative tumors had abundant MELK protein (*Figure 2G*). Given that ER/PR expression varies within the molecular HER2+ subtype, we analyzed MELK expression within this subtype. We found that MELK expression was significantly higher in ER/PR− tumors than in those with ER/PR+ status (*Figure 2H*).

An alternate categorization of breast cancers uses the expression of ER/PR and HER2. Triple-negative breast cancer (TNBC), a subtype lacking ER/PR and HER2 expression, largely overlaps with basal-like breast cancer (*Rakha et al., 2008*; *Foulkes et al., 2010*). Because this subtype-categorization

has been routinely used in the clinic for diagnosis and selection of treatment strategies, we also examined whether MELK expression correlates with this alternate subtype categorization. In two independent cohorts, the expression level of MELK is the highest in TNBC (*Figure 2I*, *Figure 2—figure supplement 1G*). Again, within the HER2+ sub-group, ER/PR− tumors have much higher MELK expression than ER/PR+ ones (*Figure 2I*). Together, these data indicate that MELK expression is highly elevated in breast tumors lacking the expression of ER and PR luminal markers.

## FoxM1 is overexpressed in BBC and regulates MELK expression

To investigate the mechanism underlying MELK overexpression in BBC, we first analyzed the copy number of MELK in breast cancer. Gene amplification of MELK occurs in both primary tumors and human breast cancer cell lines, especially in ER-negative samples (*Figure 3—figure supplement 1*). These tumors or cells with increased copy number of MELK also exhibit high level of MELK expression, suggesting that gene amplification contributes to the overexpression of MELK.

However, gene amplification of MELK occurs at low frequency, and does not explain the widespread overexpression of MELK in BBC. Recent comprehensive profiling of breast cancer suggests a role for FoxM1 activation in the transcriptional maintenance of BBC (*Cancer Genome Atlas Network, 2012*). We found that like MELK, FoxM1 is most highly expressed in the BBC or TNBC subtypes (*Figure 3A*, *Figure 3—figure supplement 2A,B*). Moreover, an extremely tight correlation between FoxM1 and MELK expression was observed in multiple large-sized cohorts (*Figure 3B*, *Figure 3—figure supplement 2C*). FoxM1 downregulation via gene silencing or a chemical inhibitor, thiostreptoin (*Hegde et al., 2011*), reduced MELK expression (*Figure 3C,D*, *Figure 3—figure supplement 2D*). Furthermore, we found that the promoter of MELK contains a putative FoxM1 binding motif (*Wierstra and Alves, 2007*), and chromatin immunoprecipitation assays using a FoxM1-specific antibody recovered a MELK promoter region that included the putative binding site (*Figure 3E*). Together these data suggest that FoxM1 is a transcription factor that is enriched in BBC and regulates MELK expression, providing a molecular mechanism underlying the overexpression of MELK in BBC.

Given that FoxM1 is a master transcription factor for many genes that are essential for mitosis (*Laoukili et al., 2005*; *Wang et al., 2005a*), our finding also suggests that MELK is a mitotic factor in BBC cells. Proteins required for mitotic progression typically accumulate during G2 and M phase, and are destroyed by ubiquitin-dependent proteolysis at the end of cytokinesis. A previous study reported that MELK is stabilized in mitosis and partially degraded upon mitotic exit in HeLa cells and *Xenopus* embryos (*Badouel et al., 2010*). We found that, in BBC cells, MELK was highly expressed during mitosis, and its protein abundance decreased dramatically when mitotic cells progressed into G1 phase (*Figure 3F*, *Figure 3—figure supplement 3*). This expression pattern of MELK, which is similar to that of Cyclin B1 and Aurora kinases, indicates that MELK is a mitotic kinase in BBC cells. Interestingly, while luminal breast cancer cells have a similar pattern of MELK expression during cell cycle, their MELK protein levels in the M phase are much lower than those of BBC cells (*Figure 3G*). The expression levels of other mitotic factors including Cyclin B1 and Aurora A are comparable between basal-like and luminal cancer cells (*Figure 3G*), suggesting that MELK may play a unique role during mitosis in BBC cells.

## Overexpression of MELK displays robust oncogenic activity

MELK was scored in our kinase library screen and is overexpressed in breast cancer, particularly in basal-like breast tumors. Therefore, we sought to further determine the potential oncogenic role of MELK. To this end, we re-engineered HMEC-DD-NeuT cells to express wild type (WT-) or myristoylated (myr-) MELK (the kinases in our initial screen were myristoylated, *Boehm et al., 2007*). While HMEC-DD-NeuT cells expressing the empty vector failed to form tumors in mice, overexpression of either WT- or myr-MELK in these cells drove tumor formation with 100% penetrance within 2 months (*Figure 4—figure supplement 1*), demonstrating that overexpression of MELK was able to confer the tumorigenicity of HMEC-DD-NeuT cells.

MELK expression strongly correlates with cell proliferation (*Venet et al., 2011*), indicating a functional role of MELK for cell growth. Indeed, we found that MELK overexpression in non-transformed HMEC-DD cells resulted in increased cell proliferation in suspension culture (*Figure 4—figue supplement 2A*). While oncogenic PIK3CA, Ras, or NeuT alone can induce colony formation of HMEC-DD cells in soft agar, two oncogenic events (e.g., PIK3CA plus NeuT), are usually required to fully transform HMEC-DD cells to form tumors in mice (*Zhao et al., 2003*, *2004*). Similar to these

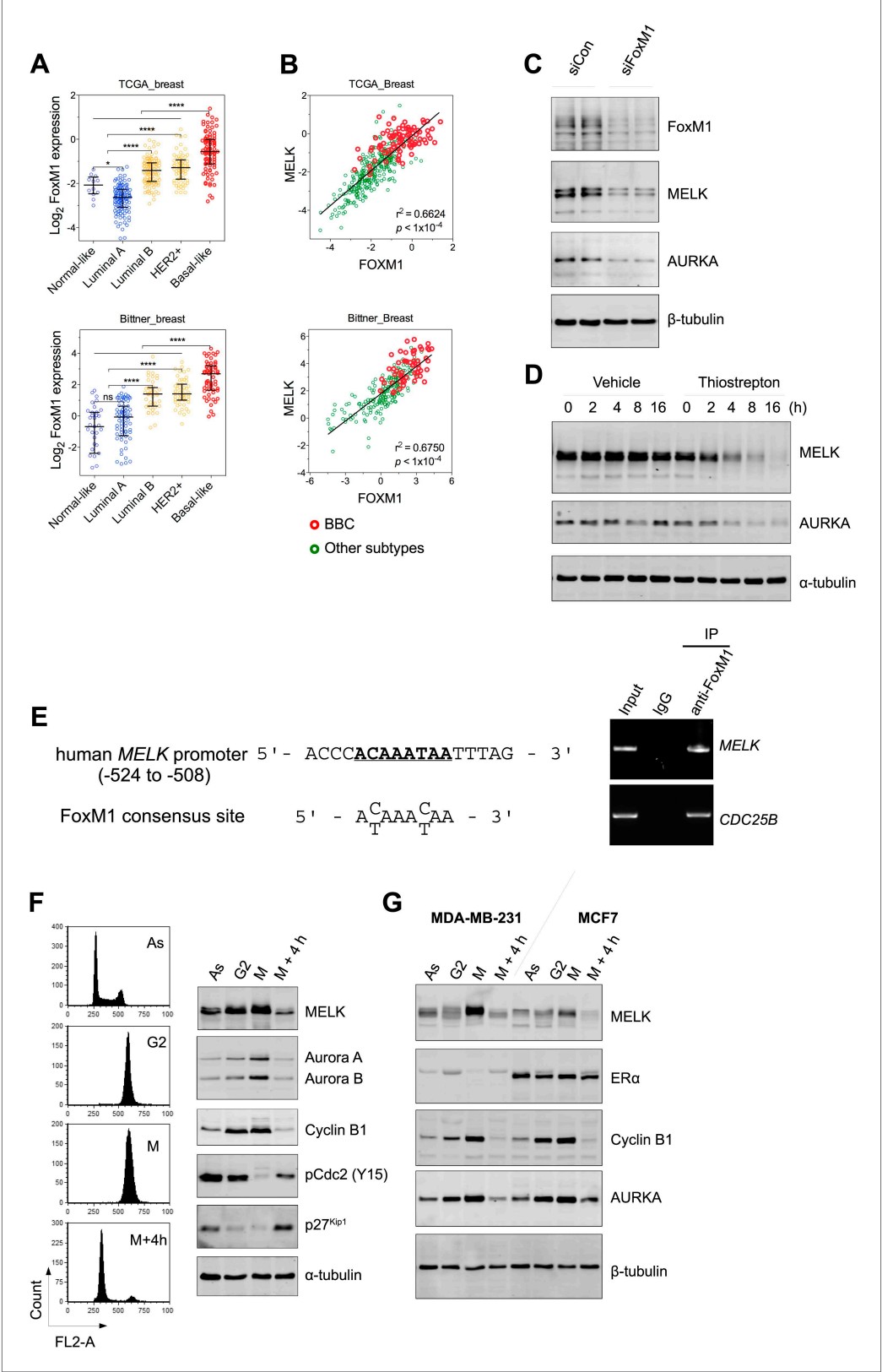

**Figure 3**. FoxM1 is overexpressed in BBC and regulates the expression of MELK. (**A**) High expression of FoxM1 in BBC. Samples in the indicated data sets were grouped into subtypes based on the PAM50 gene signature (*Parker et al., 2009*). ns, denotes not significant. *p<0.05, ****p<0.0001. (**B**) FoxM1 and MELK expression are

*Figure 3. Continued on next page*

*Figure 3. Continued*

tightly correlated. Expression of MELK was plotted against that of FoxM1. Each circle represents an individual sample of human breast carcinoma (n = 392 for TCGA dataset; n = 261 for Bittner dataset). Red and green circles represent basal-like breast tumors and all other subtypes of breast tumors, respectively. Correlation analysis was performed by GraphPad Prism. (**C**) FoxM1 knockdown suppresses MELK expression. Cells were transfected with either control siRNA or siRNA-targeting FoxM1. Lysates were harvested 3 days after transfection and subjected to immunoblotting. Aurora kinase A (AURKA), a known transcriptional target of FoxM1 (*Lefebvre et al., 2010*), was used as a positive control. (**D**) FoxM1 inhibition downregulates the expression of MELK. MDA-MB-231 cells were treated for the indicated time with vehicle or thiostrepton. Protein lysates were subjected to immunoblotting analysis of MELK and AURKA as indicated. (**E**) A putative FoxM1 binding site in the MELK promoter, and the FoxM1 consensus binding site (left). Numbers for the nucleotides are relative to the transcription start site (+1) of MELK. Chromatin immunoprecipitation assay of the MELK promoter in MDA-MB-468 cells (right). Control rabbit IgG and an antibody against FoxM1 were used. Primers for the promoter region of CDC25B were used as a positive control. (**F**) Cell cycle-dependent expression of MELK. MDA-MB-231 cells were treated with nocodazole (100 ng/ml) for 18 hr or not treated (Asynchronized, As). Nocodazole-arrested mitotic cells (M) were isolated by shake-off, and the attached cells enriched in G2 phase (G2) were harvested. A part of the mitotic cells were released into G1 phase after 4 hr of incubation (M + 4 hr). The left panel shows the flow cytometry analysis of cell cycle, and the right panel shows immunoblotting analysis of MELK and other cell cycle-specific proteins as indicated. (**G**) Expression of MELK and other mitotic factors during cell cycles in basal-like (MDA-MB-231) vs luminal (MCF7) breast cancer cells. Cell lysates were prepared as in (**F**).

The following figure supplements are available for figure 3:

**Figure supplement 1**. Gene amplification of MELK in BBC.

**Figure supplement 2**. FoxM1 is overexpressed in BBC and transcriptionally regulates MELK.

**Figure supplement 3**. MELK expression in different cell cycle of BBC cells.

---

oncogenes, over-expression of MELK alone can also promote anchorage-independent growth of HMEC-DD and MCF10A cells (*Figure 4—figure supplement 2B–E*). Likewise, wild-type MELK cooperates with a second oncogene, for example NeuT, to induce tumor formation in vivo.

In contrast to the transformation of HMEC-DD cells, one oncogenic event is sufficient to transform Rat1 rodent fibroblasts expressing p53DD (Rat1-DD) cells, as indicated by both anchorage-independent growth in vitro and tumor formation in vivo (*Ni et al., 2012*). To determine whether MELK has a transforming activity as a single event in this system, we engineered Rat1-DD cells expressing MELK (Rat1-DD-MELK), or PI3KCA H1047R (Rat1-DD-PI3KCA H1047R) as a positive control (*Figure 4A*). As expected, Rat1-DD cells transduced with an empty vector failed to grow as colonies in soft agar or to form tumors in mice. Strikingly, Rat1-DD-MELK cells displayed a robust transformed phenotype comparable to Rat1-DD-PI3KCA H1047R cells, as evidenced by both colony growth in vitro and tumor formation in vivo (*Figure 4B,C*).

To determine whether the transforming ability of MELK requires its kinase activity, we introduced catalytically inactive alleles of MELK, D150A or T167A (*Lizcano et al., 2004*; *Vulsteke et al., 2004*), into Rat-DD cells. Unlike Rat1-DD-MELK cells, both Rat1-DD-MELK-D150A and Rat1-DD-MELK-T167A cells exhibited only limited growth in soft agar or in mice (*Figure 4D–F*). Together, these studies indicate that MELK can be a potent oncogenic driver, when it is aberrantly overexpressed and that this oncogenic potential relies on its kinase activity.

## MELK is essential for the proliferation of BBC cells

Since MELK is predominantly overexpressed in basal-like breast tumors, we sought to determine whether MELK plays a role in the proliferation of BBC cells. We first analyzed a set of breast cancer cell lines that mirror the molecular subtypes of clinical tumors (*Neve et al., 2006*), and found that the expression level of MELK is much higher in the cohort of 23 BBC cell lines than in the cohort of 24 luminal breast cancer cell lines (*Figure 5A*). This is consistent with the expression pattern of MELK in primary human breast tumors. We also confirmed that the protein abundance of MELK is much higher in BBC cells compared to luminal cells (MCF7 and T47D) (*Figure 5B*). These cell lines thus, provide an excellent platform to assess potential roles of MELK in BBC.

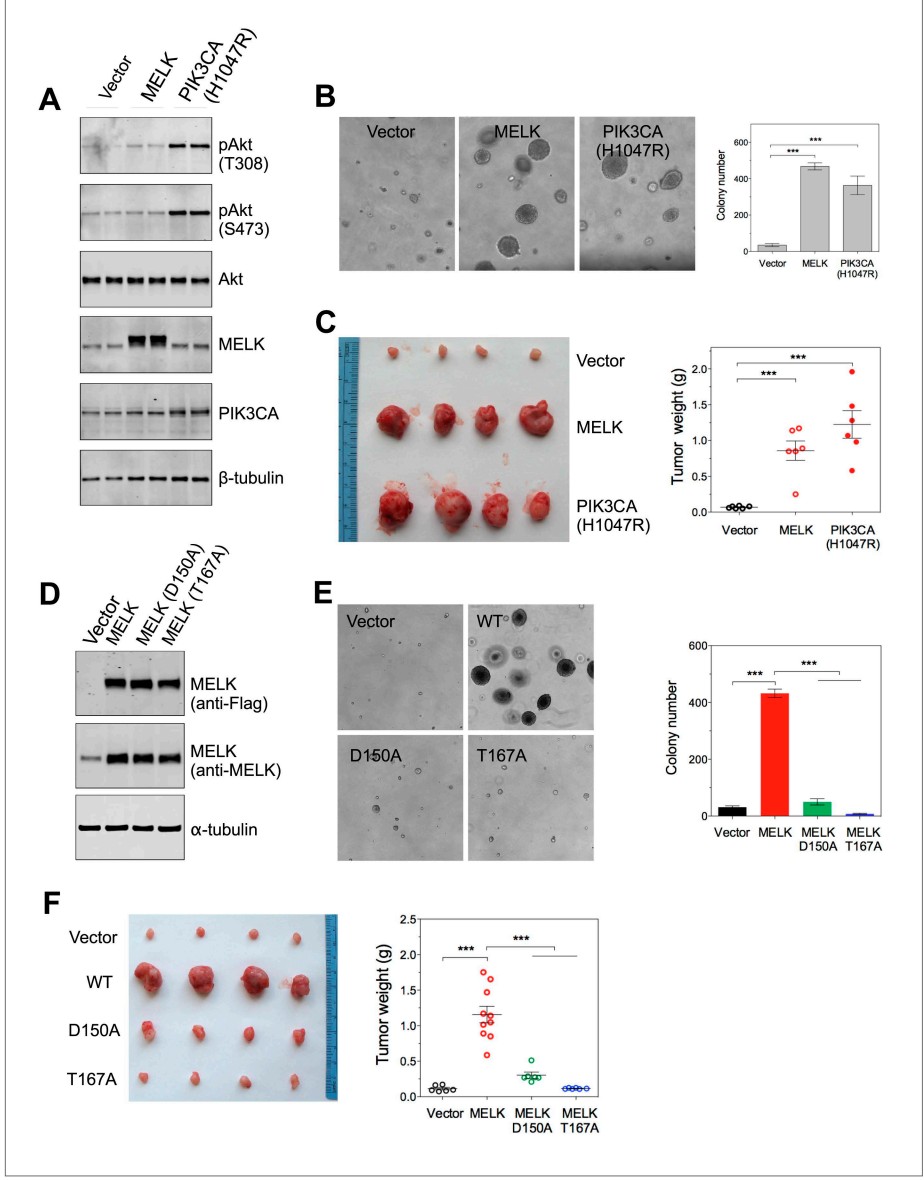

**Figure 4**. Overexpression of wild-type MELK induces oncogenic transformation. (**A**) Immunoblotting analysis of Rat1-DD cells expressing vector, wild-type (WT) allele of human MELK, or an oncogenic allele of PIK3CA (H1047R). Expression of PIK3CA (H1047R) enhances Akt phosphorylation. β-tubulin serves as a loading control. (**B**) Overexpression of MELK confers anchorage-independent growth of Rat1-DD cells. The left panel shows representative bright-field images of the anchorage-independent growth of cells expressing MELK or PIK3CA H1047R. The bar graph represents means ± SD for three experiments. (**C**) Overexpression of MELK drives Rat1-DD cells to form tumors in vivo. Representative subcutaneous tumors arising from injected Rat1-DD cells expressing MELK or PIK3CA H1047R are shown (left). The tumor weights for each group are shown as a dot chart (right). (**D**) Immunoblotting analysis of Rat1-DD cells expressing vector, WT MELK or two kinase-inactive alleles of MELK: D150A or T167A. Note that MELK is c-terminally tagged with a Flag epitope. (**E**) Rat1-DD cells expressing kinase-inactive alleles of MELK (D150A or T167A), fail to grow as colonies in soft agar. The bar graph represents means ± SD for three experiments. (**F**) Rat1-DD cells expressing kinase-inactive alleles of MELK (D150A or T167A), fail to grow as tumors in vivo. The tumor weights for each group are shown as a dot chart (right). ***p<0.001, Student's *t* test.

The following figure supplements are available for figure 4:

**Figure supplement 1**. MELK overexpression promotes tumorigenesis.

**Figure supplement 2**. MELK overexpression promotes oncogenic transformation in vitro.

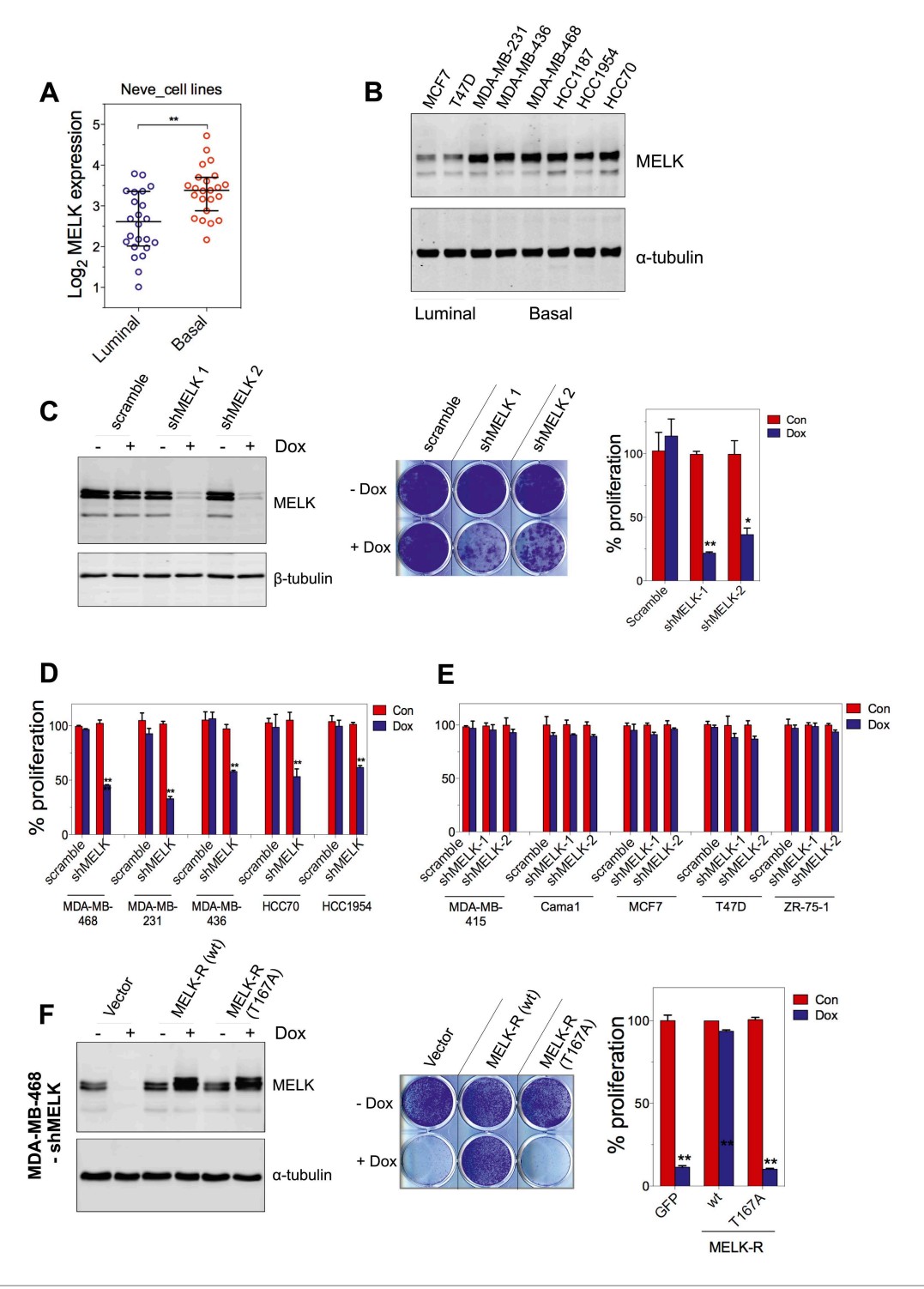

**Figure 5**. MELK is essential for the growth of basal-like breast cancer cells. (**A**) MELK expression levels are significantly higher in BBC cell lines than in luminal breast cancer cell lines. The MELK mRNA data in 23 established BBC and 24 luminal breast cancer cell lines were obtained from the Neve dataset (**Neve et al., 2006**) and are shown as a dot chart. (**B**) Immunoblotting analysis of MELK protein abundance in 6 basal-like and 2 luminal breast cancer cell lines. α-tubulin was used as a loading control. (**C**) Effects of inducible shRNA-mediated MELK silencing (tet-shMELK) in one BBC cell line, BT549. Immunoblotting analysis of MELK protein levels in the presence and absence of doxycycline is shown in the left panels. The middle and the right panels show the crystal violet staining

*Figure 5. Continued on next page*

*Figure 5. Continued*

of the plates and their respective quantification. The bar graphs indicate means ± SD for three experiments. (**D**) Effects of MELK knockdown on the proliferation of additional five BBC cell lines. Cells were treated as in (**C**). The bar graph indicates means ± SD for three experiments. (**E**) Luminal breast cancer cells are insensitive to MELK knockdown. The indicated five luminal breast cell lines were treated as in (**C**). Quantification of cell proliferation is shown (means ± SD). (**F**) WT but not a kinase-inactive allele of MELK rescues the impaired cell proliferation of BBC cells induced by MELK knockdown. The left panel shows immunoblotting analysis of MELK protein level in MDA-MB-468 cells carrying tet-shMELK, and expressing either shMELK resistant WT MELK (MELK-R) or kinase-inactive MELK (MELK-R, T167A) in the presence and absence of doxycycline. Note that the exogenous MELK is tagged with Flag epitope. The middle and the right panels show, respectively, the crystal violet staining of the plates and their respective quantification. The bar graph indicates means ± SD for three experiments. *p<0.05, **p<0.01, Student's *t* test.

The following figure supplements are available for figure 5:

**Figure supplement 1**. MELK knockdown in basal and luminal breast cancer cells.

**Figure supplement 2**. MELK knockdown does not affect the proliferation of HMECs.

**Figure supplement 3**. Generation of shMELK-resistant MELK cDNA (MELK-R).

To examine the role of MELK in the proliferation of these cells, we utilized a tetracycline-inducible gene knockdown technique in which shRNA transcription (and consequently target gene silencing) is induced upon exposure of the targeted cells to doxycycline (*Wiederschain et al., 2009*). Among the multiple inducible shRNAs that we generated to target MELK, two shRNAs were found to efficiently reduce MELK expression in cells treated with doxycycline (*Figure 5C*). We then stably introduced both shMELKs into basal or luminal breast cancer cell lines. Cell proliferation was measured upon induction of the shRNA in the presence of doxycycline. As hypothesized, MELK knockdown strongly impaired the growth of all six BBC cell lines tested, including BT549, MDA-MB-468, MDA-MB-231, MDA-MB-436, HCC70, and HCC1954 (*Figure 5C,D*, *Figure 5—figure supplement 1A,B*). In contrast, in five luminal breast cell lines, MELK knockdown with equivalent efficiency did not result in obvious inhibition on cell growth (*Figure 5E*, *Figure 5—figure supplement 1A,B*). Similarly, MELK knockdown had little effect on non-transformed HMECs, which have low level of MELK expression (*Figure 4—figure supplement 2*).

To further validate the essential role of MELK in BBC cells, we carried out rescue experiments with both WT and kinase inactive MELK. We found that the proliferation of MDA-MB-468 cells expressing shMELK was restored, when the MELK expression level in these cells was rescued by expression of a shMELK-resistant allele of MELK (MELK-R) (*Figure 5F*, *Figure 5—figure supplement 3*), confirming that the effects of shRNA are due to the specific knockdown of MELK. Notably, expression of a kinase inactive version of MELK-R, MELK-R (T167A), failed to restore cell proliferation in these cells (*Figure 5F*), indicating that kinase activity of MELK is critical for the proliferation of these BBC cells.

## Loss of MELK causes defective mitosis and cell death in BBC cells

To understand the mechanism(s) underlying the MELK function in BBC cells, we examined how inducible shRNA-mediated MELK depletion affects various cellular processes. In the presence of doxycycline, BBC cells underwent cell death indicated by increased apoptotic markers, including cleaved caspase 3, cleaved PARP, and DNA fragmentation (*Figure 6A,B*, *Figure 6—figure supplement 1A*). zVad, a pan-caspase inhibitor, was able to rescue cell death, indicating an active role of caspases in executing cell death upon MELK depletion (*Figure 6C*, *Figure 6—figure supplement 1B*). In contrast, MELK knockdown has little effect on luminal tumor cells, such as MCF7 (*Figure 6D*). To complement our RNAi-mediated MELK knockdown studies, we used a recently developed chemical inhibitor of MELK, OTSSP167 (*Chung et al., 2012*), to evaluate the functional dependency on MELK by basal and luminal breast cancer cells. Consistently, OTSSP167 induced apoptotic cell death selectively in basal breast cancer cells (*Figure 6—figure supplement 2A–C*).

The finding that MELK is a mitotic kinase in BBC cells prompted us to hypothesize that the cell death observed upon MELK inhibition might be due to an altered cell cycle progression. MELK knockdown by doxycycline induced an accumulation of cells with 4n DNA content (*Figure 6E*,

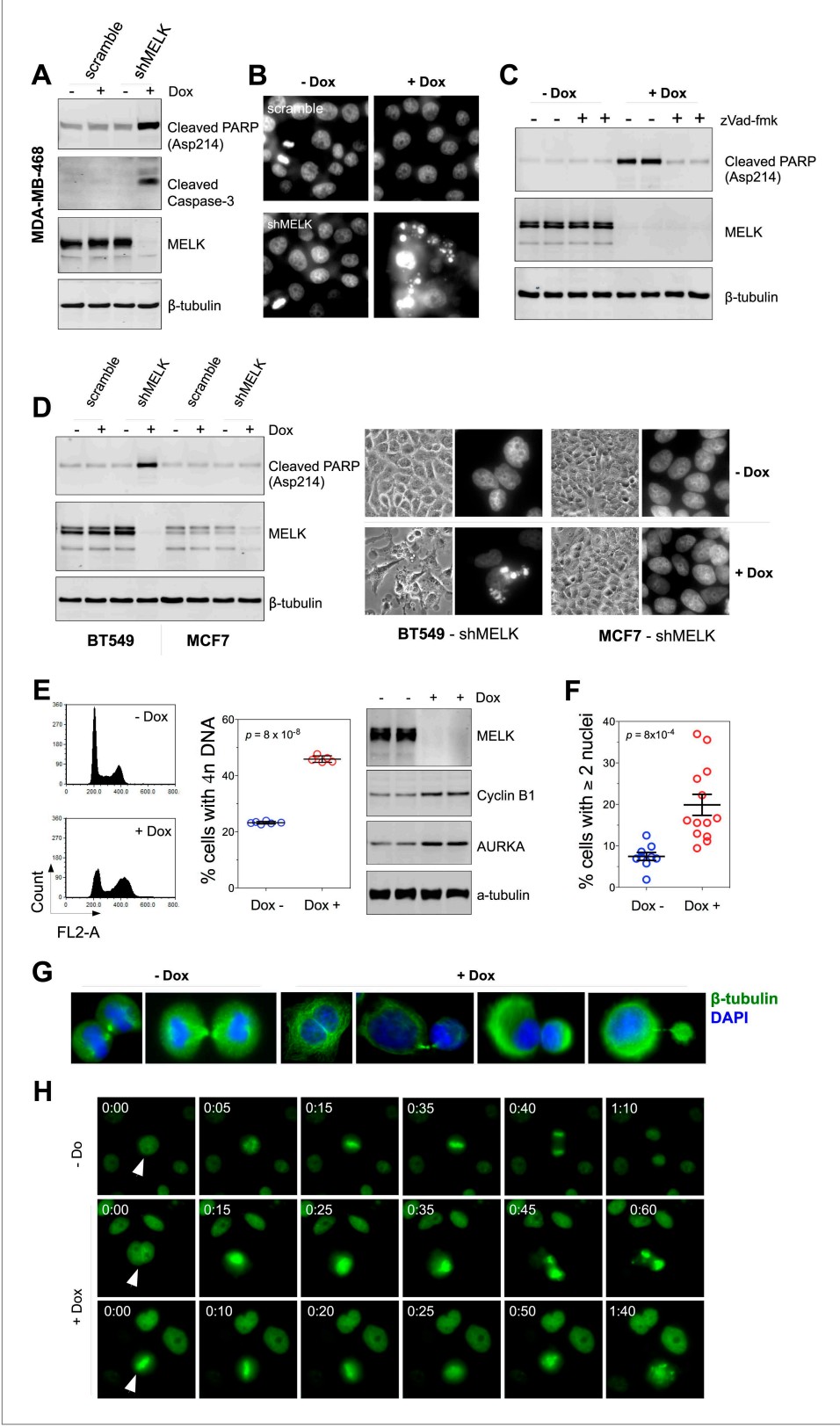

**Figure 6**. MELK downregulation induces apoptosis and impairs mitosis in BBC cells. (**A**) Immunoblotting analysis of MDA-MB-468 cells carrying a scrambled control or tet-shMELK in the presence and absence of doxycycline. Both cleaved PARP and Caspase-3 levels increased upon MELK downregulation. (**B**) MELK knockdown induces DNA
*Figure 6. Continued on next page*

*Figure 6. Continued*

fragmentation. MDA-MB-468 cells carrying scramble control or tet-shMELK were treated with or without doxycycline followed by fixation and staining with DAPI. The bright and punctate staining indicative of DNA fragmentation was only seen in cells carrying tet-shMELK in the presence of doxycycline (lower right panel). (**C**) A caspase inhibitor prevents MELK knockdown-induced cell death. MDA-MB-468 cells stably transduced with tet-shMELK were either untreated or treated with doxycycline for 4 days, and further treated with with zVad-fmk (40 μM) or vehicle during the last 2 days. Lysates from these cells were subjected to immunoblotting, with β-tubulin as a loading control. (**D**) MELK knockdown induces cell death selectively in BBC cells. The indicated cells were untreated or treated with doxycycline for 4 days followed by immunoblotting and imaging analyses. MELK knockdown induces increased level of cleaved PARP (left) and cell death (right) in BT549 but not in MCF7 cells. (**E**) MELK knockdown induces the accumulation of cells with 4n DNA content and G2/M arrest. MDA-MB-468 cells carrying tet-shMELK were treated or untreated with doxycycline for 5 days. Samples were prepared for cell cycle analysis and immunoblotting. The left panel shows representative cell cycle histograms; the middle indicates the quantification of % cells with 4n DNA content; and in the right panel, immunoblotting analysis shows that depletion of MELK increases the expression of G2/M specific proteins as indicated. (**F**) MELK knockdown induces bi- or multi-nucleated cells. MDA-MB-468 cells carrying tet-shMELK were treated or untreated with doxycycline for 4 days, followed by fixation and DAPI staining. Cells with mono-, bi-, or multi-nuclei were counted, and the data indicate % cells with two or more than two nuclei. Each circle in the histogram represents a single randomly selected field (total number of cells counted >500 for each group). The black lines indicate mean ± SEM. (**G**) MELK inhibition induces defective cell division. Fluorescent images were obtained from MDA-MB-468 cells carrying tet-shMELK as described in (**F**) stained with anti-β-tubulin (green) and DAPI (blue). (**H**) MDA-MB-468 cells stably transduced with tet-shMELK and Histone 2B-GFP were cultured in the presence or absence of doxycycline for 3 days, and then subjected to time-lapse imaging. Time is given in hours:minutes. In the absence of doxycycline, cells undergo normal mitosis (top panels). In the presence of doxycycline, binucleated cells (middle panels) and cells in metaphase (bottom panels) undergo cell death.

The following figure supplements are available for figure 6:

**Figure supplement 1**. MELK inhibition induces cell death in MDA-MB-468 cells.

**Figure supplement 2**. MELK inhibition induces cell death selectively in basal-like breast cancer cells.

**Figure supplement 3**. MELK inhibition in BBC cells induces cell death and defective mitosis.

---

*Figure 6—figure supplement 3A*), indicating an induction of G2/M arrest or failure of cytokinesis. By immunoblotting, we found that cells exposed to doxycycline exhibited an elevation of Cyclin B1 and Aurora A kinase, two markers of G2/mitosis (*Figure 6E*, *Figure 6—figure supplement 3A*). We next used microscopy to define the cell division defect in more detail. Doxycycline induced a nearly twofold increase in the percentage of cells with two or more nuclei (*Figure 6F,G*, *Figure 6—figure supplement 3B*), indicating a failure of cytokinesis. Indeed, our time-lapse microscopic analysis revealed binucleated cells forming after impaired cytokinesis (*Video 1*). Furthermore, cells in which MELK had been depleted displayed asymmetric division (*Figure 6G*), characterized by an unequal allocation of cell mass into daughter cells. Interestingly, *Caenorhabditis elegans* with mutations in the MELK homologue, PIG-1, demonstrate impaired asymmetric cell division (*Cordes et al., 2006*), supporting a critical role of MELK in the late stage of cell division.

We next used time-lapse microscopy of GFP-Histone 2B expressing cells (*Kanda et al., 1998*) to determine whether apoptosis and defective mitosis due to MELK knockdown are functionally associated. Cell death events were dramatically increased upon MELK knockdown (5 out of 235 cells in control, 151 out of 317 in doxycycline-treated cells during 10 hr of imaging). Moreover, cell death events were often preceded by division abnormalities in doxycycline-treated populations. Cells with double nuclei, which had presumably failed cytokinesis, often underwent cell death (*Figure 6H*, middle panel; *Video 2*). Some cells with an apparently normal metaphase plate were unable to progress towards anaphase, instead entering into the process of cell death directly from mitosis (*Figure 6H*, bottom panel; *Video 3*). Overall, following MELK knockdown, out of 27 examples we noted 16 failed mitoses among which 10 proceeded to cell death after the formation of metaphase plate and six gave rise to binucleated cells. By contrast, mitosis in 36 out of a total of 37 control cells appeared normal (*Figure 6H*, top panel; *Video 4*). The morphological events associated with failed cell division and ensuing cell death resembled the previous reports of the effects of inhibiting essential mitotic kinase

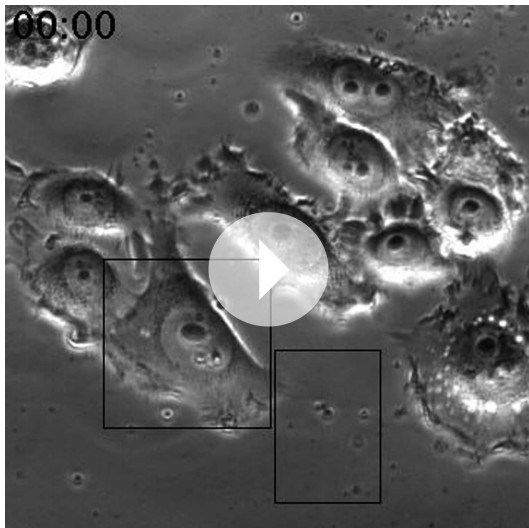

**Video 1**. This representative time-lapse video, related to **Figure 6**, shows a MDA-MB-468/tet-shMELK cell in the presence of doxycycline fails to undergo cytokinesis. The large frame indicates the initial position of the cell, and small frame its final position. Note that the cell progresses into mitosis, which ends with a double-nuclei cell following failed cytokinesis. Frame rate is five frames per second. Time is given in hours:minutes.

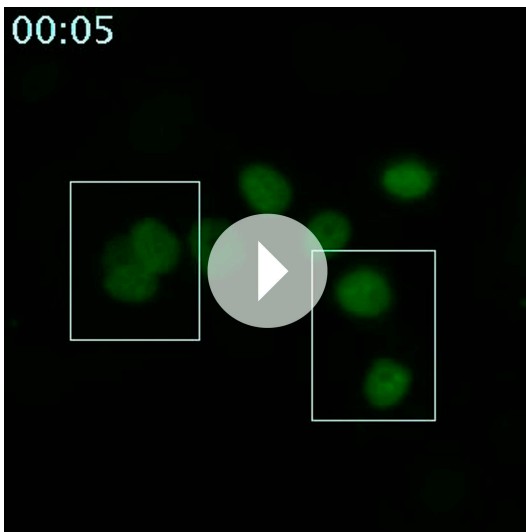

**Video 2**. This representative time-lapse video, related to **Figure 6**, shows that in the presence of doxycycline, MDA-MB-468/tet-shMELK/GFP-H2B cells with double nuclei undergo cell death. The two frames indicate two such cells. Frame rate is five frames per second. Time is given in hours:minutes.

such as Aurora B (**Keen and Taylor, 2009**). Together, these data suggest a model in which BBC cells rely on MELK for proper mitosis; inhibiting MELK in these cells causes impaired mitosis and consequent cell death.

## Therapeutic targeting of MELK in basal-like breast cancer

Since MELK is selectively required for the survival of BBC cells, we sought to determine whether MELK also supports the oncogenic growth of BBC cells using both in vitro colony formation and in vivo xenograft tumor growth assays. While MDA-MB-468 and MDA-MB-231 cells readily grew into macroscopic colonies in soft agar, MELK knockdown in these cells upon doxycycline treatment caused a nearly complete inhibition in colony formation (**Figure 7A**). To determine whether MELK is also important for BBC cells to grow as tumors in vivo, we transplanted BBC cells expressing inducible shMELK into the mammary fat pads of athymic mice to allow orthotopic tumor formation. While all recipient mice in the control group without doxycycline treatment developed tumors within 2 months, mice treated with doxycycline immediately following transplantation failed to develop tumors (**Figure 7B**), suggesting that MELK is required for the proliferation of these BBC cells in vivo. To further examine whether MELK is required for the maintenance of established tumors, we administered doxycycline to mice bearing xenograft tumors derived from basal-like or luminal breast cancer cells. Remarkably, down-regulation of MELK led to a substantial regression of tumors arising from BBC cells but had little effect on tumors derived from luminal cancer cells (**Figure 7C,D**, **Figure 7—figure supplement 1**).

To determine if the pharmacological inhibition of MELK would recapitulate the effect of MELK knockdown in xenografts, we administered OTSSP167 or vehicle control to mice that have tumors derived from basal or luminal breast cancer cell lines. While the growth of luminal breast tumors were largely unaffected by the treatment of OTSSP167, the chemical caused significant inhibition on the growth of basal breast tumors (**Figure 7E,F**). Together, these data indicate the MELK is selectively required for the oncogenic growth of BBC cells, and suggest that MELK inhibition could be an effective approach in treating basal-like breast cancer.

While MELK has a critical role in basal-like breast cancer, it is not clear whether this kinase is important for the proliferation of normal cells or tissue growth in vivo. This question is critical to the toxicity of any potential MELK-targeted therapy. To address this question, we generated mice with germ-line knockout (KO) of Melk (**Figure 7G**). Notably, Melk-deficient mice are viable and appear

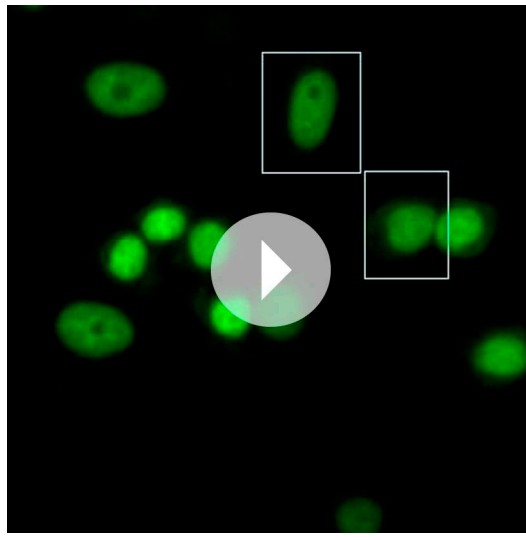

**Video 3**. This representative time-lapse video, related to **Figure 6**, shows that in the presence of doxycycline, MDA-MB-468/tet-shMELK/GFP-H2B cells undergo mitosis but ending with asymmetric cell division (in the top frame) or cell death (in the bottom frame). Frame rate is five frames per second. Time is given in hours:minutes.

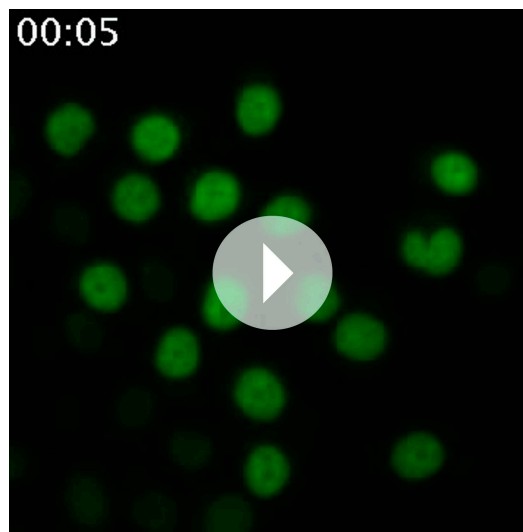

**Video 4**. This representative time-lapse video, related to **Figure 6**, shows that in the absence of doxycycline, MDA-MB-468/tet-shMELK/GFP-H2B cells demonstrate efficient mitosis. Frame rate is five frames per second. Time is given in hours: minutes.

normal without any noticeable phenotypes in the development of the embryos or adult mice. Both male and female mice are fertile and produce litters of normal size. A study by Hebbard et al. found a high activity of Melk promoter in mouse mammary progenitor cells (**Hebbard et al., 2010**). Therefore, we anticipated an impairment of mammary gland development in Melk KO mice. However, mammary glands from mice with Melk KO appear normal in both morphology and function (e.g., lactation).

Since bone marrow toxicity is a major side effect of most anti-cancer therapies and, in particular, of those drugs targeting mitotic kinases/machinery, we characterized the immune system in Melk-deficient mice. We isolated cells from bone marrow, spleen and thymus, and analyzed immune cell populations including monocytes, neutrophils, B and T cells. We also analyzed the hematopoietic stem cells and various progenitor cells in the bone marrow. In both cases, virtually no differences were found between wild-type and KO mice (**Figure 7H,I**). Together, these data suggest that Melk is not essential for normal development and physiological functions in mice, providing compelling evidence for MELK as a highly selective target for therapeutic intervention of basal-like breast cancer.

## Discussion

Patients with basal-like breast cancer remain faced with limited treatment options due to the aggressive nature of the disease and the current lack of suitable molecular targets for therapeutic intervention. In this study, we report that MELK, a novel oncogenic kinase that emerged from an unbiased, in vivo tumorigenesis screen, may indeed be a therapeutic target in this tumor type. In a comprehensive analysis of databases with multiple cohorts of breast cancer, we find MELK to be highly overexpressed in breast cancer lacking the expression of ER/PR, including basal-like breast cancer. Remarkably, overexpression of wild-type MELK induces robust oncogenic transformation both in vitro and in vivo with a transforming potency comparable to that of the highly oncogenic mutant allele of PIK3CA. Even more striking is the finding that only basal-like, but not luminal breast cancer cells, depend on MELK for proliferation. In addition, the dispensable nature of Melk in normal development and hematopoiesis in mice underlines its selective role in BBC. Notably, the kinase activity of MELK is required for its transforming activity as well as for the survival and proliferation of BBC cells. Thus, MELK is potentially a novel oncogenic driver of basal-like breast carcinoma and a promising target for small molecule-based therapeutic intervention.

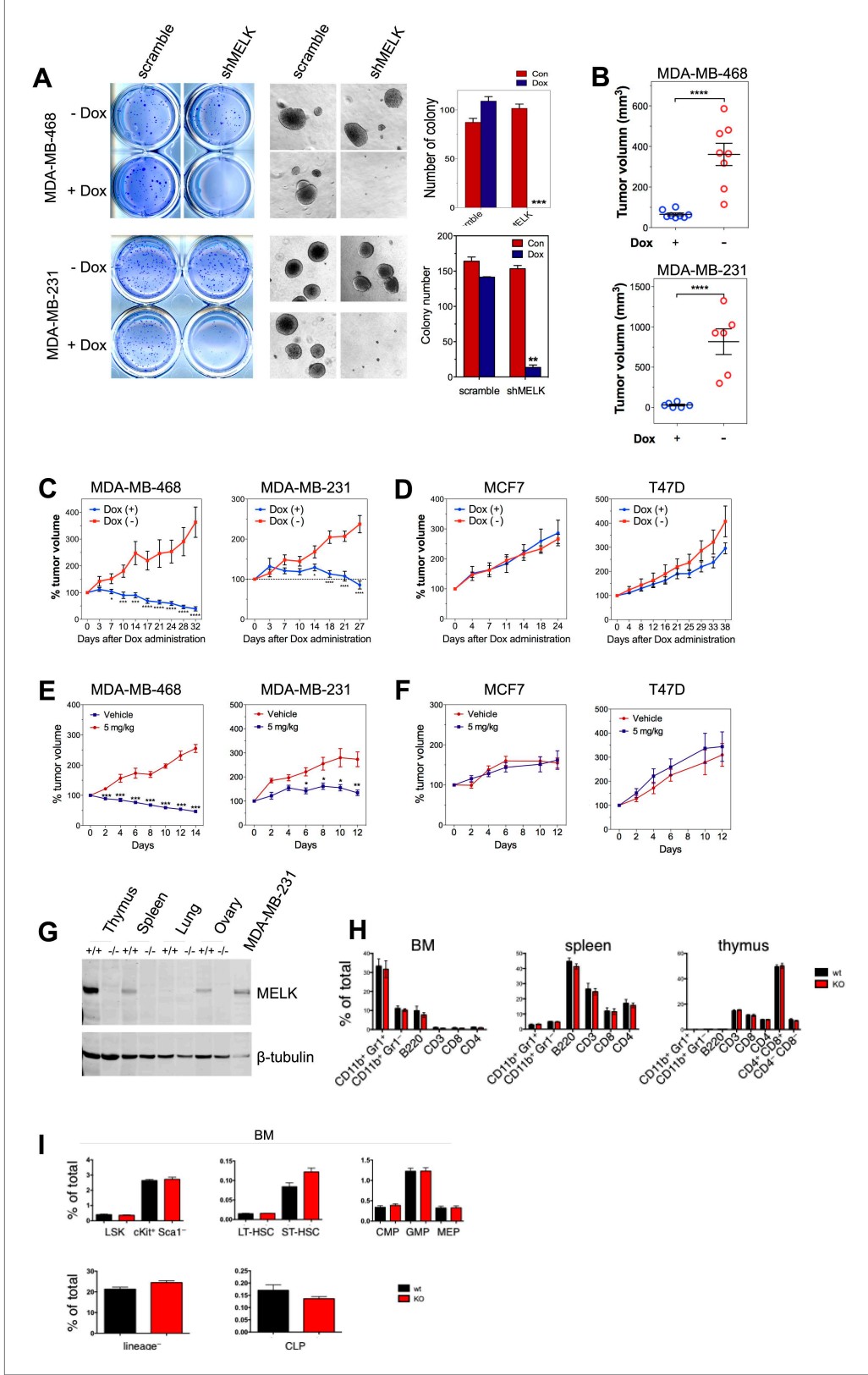

**Figure 7**. MELK is essential to sustain the oncogenic growth of BBC Cells. (**A**) Effects of MELK knockdown on anchorage-independent growth of BBC cells in soft agar. The left and middle panels show crystal violet staining, and bright-field images of the colonies respectively. The bar graphs indicate the means ± SD for three

*Figure 7. Continued*

experiments. (**B**) Effects of MELK knockdown on the growth of BBC cells in vivo. MDA-MB-468 and MDA-MB-231 cells carrying tet-shMELK were orthotopically implanted into the mammary fat pads of nude mice. The recipient mice were divided into two groups: one group of mice was given doxycycline-supplemented drinking water on the second day of injection for the duration of the experiment, while the other group of mice was maintained without doxycycline. The histogram indicates tumor volume measured 7 weeks after treatment. Data are means ± SEM (n ≥ 6). (**C** and **D**) Effects of MELK knockdown on established tumors arising from implantation of basal (**C**) or luminal (**D**) breast cancer cells. Mice bearing orthotopic tumors arising from the indicated cells carrying tet-shMELK were divided into two groups, with one group of mice receiving doxycycline, and the other maintained without doxycycline. Tumor volumes were measured on the indicated days after the administration of doxycyline. Data are means ± SEM (n ≥ 8). (**E** and **F**) Effects of MELK inhibition on tumor growth. Mice with tumors developed from basal (**E**) or luminal (**F**) breast cancer cells, were treated once daily with vehicle (0.5% methycellulose) or OTSSP167 (5 mg/kg). Tumor volumes were measured on the indicated days. Data are means ± SEM (n ≥ 8). (**G**) Knocking out *Melk* in mice. Indicated tissues were harvested from *wild type* or *Melk*$^{-/-}$ adult mice, and homogenized in RIPA lysis buffer. Lysates were subjected to immunoblotting. Total lysate of human breast cancer cell line MDA-MB-231 was used as a control. (**H**) Loss of Melk has no obvious impact on the development of immune system. Cells were isolated from bone marrow (BM), spleen and thymus, and subjected to flow cytometric analysis. Note that CD11b+/Gr1+ is a marker for neutrophils, CD11b+/Gr1− for monocytes, B220 for B cells, CD3, CD4 and CD8 for T cells. (**I**) Bone marrow was collected from *wild type* (wt) and *Melk*$^{-/-}$ (KO) mice and stained for the indicated cell populations. LSK: Lin$^-$Sca1$^+$ ckit$^+$; LT-HSC(long-term hematopoietic stem cells): LSK CD150$^+$CD48$^-$; ST-HSC(short-term hemato-poietic stem cells): LSK CD150$^-$CD48$^-$; CMP(Common myeloid progenitor): Lin$^-$cKit$^+$Sca1$^-$IL7Ra$^-$CD34$^+$FcRg$^-$; GMP(Granulocyte-macrophage progenitors): Lin$^-$cKit$^+$Sca1$^-$IL7Ra$^-$CD34$^+$FcRg$^+$; MEP(Megakaryocyte-erythrocyte progenitors): Lin$^-$cKit$^+$Sca1$^-$IL7Ra$^-$CD34$^-$FcRg$^-$; CLP(Common lymphoid progenitors): Lin$^-$cKit$^{mid}$Sca1$^{mid}$IL7Ra$^+$. *p<0.05, **p<0.01, ***p<0.001, ****p<0.0001, Student's *t* test.

The following figure supplements are available for figure 7:

**Figure supplement 1**. Efficient conditional MELK knockdown in vivo.

Our data point to a potential role for MELK as a marker in predicting disease outcome. In multiple independent breast cancer cohorts analyzed, we found a strong association of high expression levels of MELK with a higher grade of malignancy and an unfavorable prognosis regardless of the treatment modality. While high MELK expression seems to be a unique phenomenon for BBC in breast cancer, MELK overexpression has been associated with tumor aggressiveness and poor outcome in a number of other cancer types, including glioblastoma (*Nakano et al., 2008*), astrocytoma (*Marie et al., 2008*), and prostate cancer (*Kuner et al., 2013*). The prognostic feature of MELK expression is likely due to its correlation with cell proliferation. In fact, MELK and other proliferation-related genes are major components of multi-gene signature for predicting disease outcome. For example, a recent study developed a cell proliferation signature that consists of the top 1% genes whose expression is most positively correlated with that of proliferating cell nuclear antigen (PCNA). The authors found that adjusting breast cancer expression data for this cell proliferation signature causes a dramatic reduction in outcome association of most published breast cancer signatures (*Venet et al., 2011*). Notably, MELK expression strongly correlates with cell proliferation, and in fact is one of the top-ranking signature genes of cell proliferation that correlate with PCNA expression (*Venet et al., 2011*).

Previous studies demonstrated that, while MELK is a member of the AMPK family, it is not activated via phosphorylation by the tumor suppressor kinase LKB1 (*Lizcano et al., 2004*). Recombinant MELK expressed in bacteria is catalytically active (*Davezac et al., 2002*; *Lizcano et al., 2004*; *Beullens et al., 2005*). Consistent with these findings, overexpression of wild-type MELK readily drives transformation in vitro and in vivo. This behavior is similar to that of other established proto-oncoproteins, such ERBB2 (*Di Fiore et al., 1987*; *Hudziak et al., 1987*), and Aurora A kinase (*Bischoff et al., 1998*; *Zhou et al., 1998*), the transforming activity of which is driven by overexpression of the wild-type protein. While a number of substrates have been proposed for MELK, such as Bcl-G (*Lin et al., 2007*), CDC25B (*Davezac et al., 2002*), p53 (*Seong and Ha, 2012*), and PDK1 (*Seong et al., 2012*), the substrates that mediate the oncogenic activity of MELK in breast cancer remain to be identified.

An intriguing question is how the selective overexpression of MELK is achieved in BBC. Our finding that the mitotic transcription factor FoxM1 (*Laoukili et al., 2005*; *Wang et al., 2005a*) plays a major role in regulating MELK expression has shed some light on this enigma. Notably, the expression levels

of FoxM1 and MELK demonstrate a striking correlation across all breast cancer samples and subtypes examined. Like MELK, FoxM1 is significantly overexpressed in BBC. Consistent with our results, FoxM1 was recently proposed as a transcriptional driver of proliferation-associated genes in BBC (*Cancer Genome Atlas Network, 2012*).

However, why MELK is selectively required for cell division in BBC cells, but not in other types of breast cancer or normal cells, remains an open question. MELK was not observed as a hit in systematic screens for essential cell division proteins in HeLa cells (*Kittler et al., 2004*). Both *C. elegans* and mice are tolerant of mutation or deletion of MELK ortholog (*Cordes et al., 2006*; *Figure 7G–I*). However, MELK can be essential in some circumstances. It is expressed in early frog embryos, where it seems to play some role in cell division (*Le Page et al., 2011*), and we observed it accumulating in dividing cells (*Figure 3F*), and playing an important role during cell division in BBC cells (*Figure 6*). To reconcile these apparently disparate findings, we propose that one or more MELK-related kinase is required for cell division in many, if not all vertebrate cells. In BBC cells, MELK must play this role uniquely and is selectively overexpressed, perhaps because redundant kinases are down-regulated. In other cells, MELK may function during division, but it is not essential due to redundancy with related kinases. Consistent with this hypothesis, the MELK-related kinase AMPK was recently shown to play a role in mitosis (*Vazquez-Martin et al., 2009*). Perhaps AMPK, or other kinases in the same family, can substitute for MELK in some cells, but not in BBC cells, which seem to have become addicted to MELK for proper execution of cell division. Determining the precise function of MELK in cell division, and the reason this function is selectively required in BBC cells, will require further analysis. Nevertheless, our studies firmly establish MELK as a molecular target for the treatment of BBC. Unlike other mitotic factors like Aurora A, Aurora B, and PLK1, that are normally essential, MELK presents a unique mitotic kinase that is only required by a subset of cancer cells, and is therefore an excellent therapeutic target.

In summary, recent comprehensive characterization of basal-like breast cancer demonstrates that this subtype of disease has high genetic heterogeneity, but lacks commonly occurring genetic alterations, with the exception of the frequent inactivation of p53 (*Cancer Genome Atlas Network, 2012*). In contrast, the relative uniform overexpression of MELK in basal-like breast cancer makes it a potential common target in an otherwise heterogeneous disease. Thus our data on MELK provide important information for guiding the development of targeted therapies in basal-like breast cancer.

# Materials and methods

## Plasmids

The human MELK was amplified using the template DNA deposited in the described kinase library, and cloned into pWZL retroviral vector (*Zhao et al., 2003*), in which target gene expression is driven by the long terminal repeat of Moloney murine leukemia virus. The MELK mutants (D150A, T167A, or shMELK-resistant MELK with silent mutations) were generated via Quickchange XL Site-directed Mutagenesis (Stratagene, La Jolla, CA). Primers were listed in *Supplementary file 2*.

To construct a tetracycline-inducible gene expression system, GFP or mutated MELK was amplified using the primers listed in *Supplementary file 2*. The PCR products were digested with AgeI and PacI, and ligated with digested pLKO-TREX (*Wee et al., 2008*).

To construct pWzl-H2B-GFP, human Histone 2B was amplified using the genomic DNA of HEK293T cells as templates. Primers for cloning were listed in *Supplementary file 2*. PCR products following digestion with BamHI and XhoI were ligated with digested pWzl-GFP.

To generate pLKO-tet-on-shRNAs targeting human MELK, oligonucleotides were designed and synthesized (IDT, Coralville, Iowa). Following annealination, double-stranded oligonucleotides were directly ligated with pLKO vector that was digested with AgeI and EcoRI. The sequences for scramble, shMELK1, shMELK2 are listed in *Supplementary file 2*.

Retroviruses were generated by transfecting HEK293T cells with pWzl plasmids and packaging DNA. Typically 1.6 μg pWzl DNA, 1.2 μg pCG-VSVG and 1.2 μg pCG-gap/pol, 12 μl lipid of Metafectene Pro (Biontex, Martinsried, Germany) were used; DNA and lipid were diluted in 300 μl PBS respectively and mixed; and following 15 min of incubation, they were added to one 6-cm dish that was seeded with 3 million HEK293T cells 1 day earlier. Viral supernatant was collected 48 hr and 72 hr after transfection. After the supernatant was filtered through 0.45-μm membrane, it was added to target cells in the presence of 8 μg/ml polybrene (Millipore, Billerica, MA). Lentiviruses were generated with a similar approach with the exception of HEK293T cells that were transfected with 2 μg pLKO DNA, 1.5 μg pCMV-dR8.91,

and 0.5 µg pMD2-VSVG. Cells were selected with antibiotics starting 72 hr after initial infection. Puromycin and blasticidin were used at the final concentrations of 1.5 µg/ml and 4 µg/ml respectively.

## Cell culture

Human mammary epithelial cells (HMECs) were maintained in DMEM/F-12 supplemented with EGF (10 ng/ml), insulin (10 µg/ml), and hydrocortisone (0.5 µg/ml) under 5% $CO_2$ and 37°C. Rat1 and HEK293T cells were maintained in DMEM supplemented with 10% FBS (Invitrogen, Carlsbad, CA). All breast cancer cell lines (MCF7, T47D, MDA-MB-468, MDA-MB-231, MDA-MB-436, HCC1197, BT549) were cultured in RPMI 1640 medium supplemented with 10% FBS. For cells stably introduced with tetracyclin-inducible genes/shRNAs, Tet-approved FBS (Clontech, Mountain View, CA) was used.

## Cell proliferation assay

Typically, breast cancer cells were seeded in 12-well plates ($1–2 \times 10^4$) in 1 ml medium. On the next day, wells were added with 110 µl medium without or with 1 µg/ml doxycycline (to reach a final concentration of 100 ng/ml), which was repeated every 2 days. 6 days after the initial treatment, cells were fixed with formaldehyde, and stained with crystal violet (0.05%, wt/vol), a chromatin-binding cytochemical stain. The plates were washed extensively, and imaged with a flatbed scanner. For quantification of the staining, 1 ml 10% acetic acid was added to each well to extract the dye. The absorbance was measured at 590 nm with 750 nm as a reference.

## Colony formation assay

The assays were typically performed in a 12-well plate unless otherwise mentioned. Cells were suspended in medium containing 0.3% agar and plated onto a layer of 0.6% agar (for each well, 4000 cell in 800 µl medium, 1 ml bottom agar). The wells were added with medium (without or with 100 ng/ml doxcycycline) on the next day. 3 weeks after seeding, the colonies were fixed with formaldehyde and imaged. The number of colonies in each well was quantified using ImageJ (National Institutes of Health).

## Tumor xenograft studies

All xenograft studies were conducted in accordance with the animal use guidelines from the National Institutes of Health and with protocols approved by the Dana-Farber Cancer Institute Animal Care and Use Committee. The recipient mice used were NCR-nude (CrTac:NCr-Foxn1nu, Taconic, Hudson, NY). Cells were resuspended in 40% of Matrigel-Basement Membrane Matrix, LDEV-free (BD Biosciences, San Jose, CA) and sit on ice until injection. For transplanting human cell lines, mice were γ-irradiated with a single dose of 400 rads on the same day of injection. Mice were anesthesized by inhalation of isoflurane, and were injected with 150 µl cells ($5 \times 10^6$) per site. Tumors were measured in two dimensions by a caliper. Tumor volume was calculated using the formula: V = 0.5 × length × width × width. All xenograft data are presented as mean ± SEM. Comparison between groups of treatment were conducted using two-tailed Student's $t$ test. Calculations were performed using either Openoffice or GraphPad Prism version 5.0b.

For tumorigenesis study, $5 \times 10^6$ HMEC cells were injected into the mammary fat pad, and $5 \times 10^6$ Rat1 cells subcutaneously. Tumor growth was monitored twice a week. Rat1 xenografts were harvested 3 weeks after injection.

To study the impact of MELK knockdown on tumor growth, mice were randomly sorted into groups on the second day of injection, and were untreated or treated with doxycycline (2 mg/ml in 5% dextrose in drinking water, refreshed twice a week) for the duration of the study. Tumor was measured twice a week.

To study the roles of MELK in tumor maintenance, mice with established tumors ($\geq 200$ mm$^3$) derived from orthotopic injections of MDA-MB-231, or MDA-MB-468, or MCF-7, or T47D cells were randomly sorted into two groups, with one group receiving doxcycline in drinking water. Tumors were calipered twice per week to monitor the effect of MELK knockdown on tumor growth.

## Time-lapse imaging

Time-lapse imaging was performed on a Nikon Ti motorized inverted microscope, which was equipped with a perfect focus system and a humidified incubation chamber (37°C, 5% $CO_2$) (Nikon Imaging Center, Harvard Medical School). Cells stably expressing H2B-GFP were pre-seeded in 24-well glass-bottom plate, and either untreated or treated with doxcyline (100 ng/ml final). Images were captured every 5 min with a 20× objective lens, and a Hamamatsu ORCA-AG cooled CCD camera. Images were analyzed using ImageJ (National Institutes of Health).

## Immunofluorescence analysis

Cells were seeded on No. 1.5 coverslips (12 mm round) that were pre-placed into 24-well plates. Upon harvest, cells were fixed with 4% formaldehyde for 10 min. After washing, cells were permeablized with 0.1% Trition X-100 for 10 min. Cells were then washed and blocked with 1% bovine serum for 30 min before incubated with primary antibody (anti-β-tubulin, #2128; Cell Signaling Technology, Beverly, MA) prepared in PBS containing 1% bovine serum albumin. After overnight incubated at 4°C, the samples were washed and incubated with Alexa 488-conjugated secondary antibody (Invitrogen) for 1 hr at room temperature. After extensive washing, the samples were dried and mounted with ProLong Antifade reagent (Invitrogen). The images were acquired with a Nikon 80i upright microscope at the Nikon Imaging Center (Harvard Medical School), which is equipped with a Hamamatsu C8484-03 monochrome camera. ImageJ was used for analysis of the images, which includes merging channels with different colors and cropping.

## Immunoblotting

Cells were lysed with RIPA buffer (25 mM Tris, pH 7.4, 150 mM NaCl, 1% Nonidet P-40, 0.5% sodium deoxycholate, and 0.1% sodium dodecyl sulfate) supplemented with protease inhibitors cocktail (Roche) and phosphatase inhibitors cocktail (Thermo Scientific, Waltham, MA). Cleared lysates were analyzed for protein concentration using BCA kit (Thermo Scientific). Equal amount of protein (10–20 μg) was resolved on SDS-PAGE, and was subsequently transferred onto a nitrocellulose or polyvinylidene difluoride membrane. The membrane was blocked with 5% non-fat milk and was then incubated with primary antibodies overnight at 4°C. After washing, the membrane was incubated with fluorophore-conjugated secondary antibodies for 1 hr at room temperature. The membrane was then washed and scanned with an Odyssey Infrared scanner (Li-Cor Biosciences, Lincoln, NE). Primary antibodies used in this study include anti-MELK, anti-α-tubulin (Abcam, Cambridge, MA), anti-cyclin B1 (Millipore), anti-Vinculin (Sigma, St. Louis, MO), anti-FoxM1 (Santa Cruz, Dalla, TX), anti-β-tubulin, anti-phopho-Akt (S473), anti-phospho-Akt (T308), anti-total Akt, anti-Flag, anti-cleaved PARP (Asp214), anti-cleaved Caspase-3, anti-AURKA, anti-AURKB, anti-p27, anti-Estrogen Receptor α (all from Cell Signaling Technology). Secondary antibodies used were IRDye700-conjugated anti-rabbit IgG and IRDye800-conjugated anti-mouse IgG (Rockland, Gilbertsville, PA).

Primary human breast cancer samples were obtained from the Dana-Farber Cancer Institute with patients' consent and institutional review board approval. These samples were deidentified and are not considered human subject research. Samples were homogenized in RIPA buffer supplemented with protease/phosphatease inhibitors using Bullet blender (Next advance, Averill Park, NY). After clearing, tissue lysates were subjected for protein concentration determination. 20 micrograms of lysates were used for immunoblotting.

## Generation of Melk knockout mice

Mouse embryonic stem cells with one allele of Melk inserted with lacZ and neomycin-resistance genes between exon 2 and 3 were obtained from the Knockout Mouse Project (KOMP; ID: CSD33136). Cells with normal karyotype were injected into blastocysts isolated from C57BL/6 mice. The procedure of injection was performed at the Transgenic Core Facility, Brigham and Women's Hospital (Boston, MA). Germline transmission was subsequently observed, and further cross was made to generate Melk homozygous knockout mice. Melk knockout was confirmed by long-rang PCR, qPCR, and immunoblotting.

## Flow cytometry

Cells were isolated from bone marrow, spleen, and thymus of mice, and stained with the following antibodies: B220 (APC; BD Pharmingen), cKit (PE-Cy7; BioLegend, San Diego, CA), CD3 (PE-Cy7; BD Bioscience), CD4 (APC-H7; BD Pharmingen), CD8 (ECD; Beckman Coulter), CD11b (PE; BD Bioscience), CD16/32 (PE; eBioscience, San Diego, CA), CD34 (FITC; BD Pharmingen), CD45.2 (PerCP-Cy5.5; BD Pharmingen), CD48 (APC-Cy7; BD Pharmingen), CD127 (ECD; BD Pharmingen), CD150 (PerCP-Cy5.5; BioLegend), Gr1 (APC-Alexa700; BD Bioscience), Lineage Cocktail (APC; BD Pharmingen), Sca1 (Brilliant Violet 421; BioLegend). Dead cells were excluded using either DAPI or Vivid-Aqua (Invitrogen) staining. All data acquisition was performed on a LSRII (BD) flow cytometer, and results were analyzed using FlowJo v.8.8.7 (TreeStar).

## Cell cycle analysis

Cells were harvested by trypsinization, and repeatedly pipetted into single-cell suspension. After centrifugation, cells were fixed by adding 70% ethanol (−20°C) dropwise while vortexing. Cells were then

stained with propidium iodide (50 µg/ml, Sigma) solution containing 50 µg/ml DNase-free RNase A (Sigma) and 0.5% bovine serum albumin (BSA). After 30 min of incubation, the samples were washed and resuspended in 0.5% BSA. The analysis was performed on a LSRFortessa (BD Biosciences) at the DFCI Flow Cytometry Core Facility. Single cells were gated via plotting FL3-A to FL3-H to exclude cell debris and doublets. At least 10,000 single cells were collected for each sample.

## Chromatin immunoprecipitation

Chromatin immunoprecipitation was performed as previously described (*Lee et al., 2006*). Upon harvest, medium in cell culture dishes was added with 16% formaldehyde (Electron Microscopy Sciences, Hatfield, PA) to reach a final concentration of 1%, and quenched with glycine (125 mM final, 5 min incubation) after incubation at room temperature for 10 min. Cells were harvested by scrapping into cold PBS, and centrifuged. Cell pellets were lysed with LB1 (50 mM HEPES, pH 7.5, 140 mM NaCl, 1 mM EDTA, 10% glycerol, 0.5% NP-40, 0.25% Triton-X-100), then after centrifugation with LB2 (10 mM Tris–HCl pH 8.0, 200 mM NaCl, 1 mM EDTA, 0.5 mM EGTA), and again after centrifugation resuspended in LB3 (10 mM Tris–HCl pH 8.0, 100 mM NaCl, 1 mM EDTA, 0.5 mM EGTA, 0.1% Na-Deoxycholate, 0.5% N-lauroylsacosine). Samples were sonicated using a Q800R DNA Shearing Sonicator (Qsonica, Newtown, CT) at 50% amplitude for 10 min with a pulse of 30 s on and 30 s off. Samples were then supplemented with 10% Triton-X 100 to a final concentration of 1%, and centrifuged at 20,000×*g* for 10 min at 4°C. The cleared lysates were used for the following immunoprecipitation, with 50 µl of lysate saved as input.

Protein G-conjugated Dynabeads (Invitrogen) were washed with block solution (0.5% bovine serum albumin in PBS) and incubated overnight with 5 µg anti-FoxM1 (SC-502, Santa Cruz Biotechnology), or 5 µg rabbit IgG in block solution, and on the next day washed three times with block solution. Cell lysates were incubated with the antibody/magnetic bead, rotating at 4°C overnight. On the next day, the beads were collected with magnetic stand, and washed six times with RIPA buffer (50 mM HEPES pH 7.6, 500 mM LiCl, 1 mM EDTA, 1% NP-40, 0.7% Na-deoxycholate). After a single wash with Tris-EDTA buffer containing 50 mM NaCl, samples were resupended with elution buffer (50 mM Tris–HCl pH 8.0, 10 mM EDTA, 1% SDS) for incubation at 65°C overnight. Also, the 50 µl input was mixed with 150 µl elution buffer and incubated at 65°C overnight for reverse crosslinking. On the next day, RNase A was added to the samples (0.2 µg/ml final), followed by incubation for 1 hr at 37°C. Samples were then treated with Proteinase K (0.2 µg/ml final) and incubated at 56°C for 1 hr. DNA were purified with a QIAquick PCR purification kit (Qiagen), and eluted with 30 µl water. PCR was performed using Quick-Load Taq 2X Master Mix (New England BioLabs, Beverly, MA), using primers listed in *Supplementary file 2*.

## qRT-pcr analysis

Total RNA was extracted from cultured cells with RNeasy Mini kit (Qiagen), with the use of QIAshredder spin column for homogenization and an on-column DNase digestion. 2 µg of the total RNA was reversely transcribed using a High Capacity RNA-to-cDNA Kit (Applied Biosystems, Foster City, CA). cDNA were analyzed quantitatively using Power SYBR Green PCR Master Mix (Applied Biosystems) on an ABI7300 Real-time PCR system. Primers used were listed in *Supplementary file 2*. Cycling conditions were 95°C for 15 min, 40 cycles of 15 s at 94°C, 30 s at 55°C and 30 s at 72°C. Ct values were generated using the default analysis settings. $\Delta CT$ was defined as $Ct_{gene\ of\ interest} - Ct_{\beta\text{-actin}}$. $\Delta\Delta CT$ was defined as $\Delta Ct_{treated\ sample} - Ct_{control\ sample}$. Relative quantification (RQ) was calculated as $2^{-\Delta\Delta CT}$. Statistical analysis was performed by Student's *t* test.

## Analysis of gene expression

Gene expression data were downloaded from Oncomine (*Rhodes et al., 2004*). Information of the clinical data sets is listed in *Supplementary file 1*. Analyses and figures were made in GraphPad Prism. In dot plot graphs, each dot indicates an individual sample, with results expressed as median with interquartile range.

## Survival analysis

Independent cohorts of breast cancer patients with overall survival or metastasis-free survival data available were examined. Information of the cohorts is listed in *Supplementary file 1*. Data of MELK expression and associated survival were downloaded from Oncomine (*Rhodes et al., 2004*). For each cohort, patients were divided into top 60% 'MELK high' and bottom 40% 'MELK low' groups based on the expression of MELK. Kaplan–Meier curves, as well as the log-rank (Mantel–Cox) test and the hazard ratio were analyzed by GraphPad Prism.

## Statistical analysis

Two-tailed Student's *t* test and ANOVA (Analysis of Variance) were used for differential comparison between two groups and among three groups, respectively. Survival and correlation analysis were performed in GraphPad Prism.

## Acknowledgements

We thank Drs TM Roberts and DM Livingston for scientific discussions. We thank the Nikon Imaging Center at Harvard Medical School, DFCI Flow Cytometry Core Facility, and the Transgenic Core Facility at the Brigham and Women's Hospital for technical assistance and the use of instruments. This work was supported by Friends of Dana-Farber Cancer Institute (YW), DFCI/Accelerator Fund (NSG, JJZ), and NIH grants (JJZ).

## Additional information

### Competing interests

YW: Patent pending on the use of MELK kinase inhibitors in the treatment of breast cancer, PCT/US2014/010724. Y-ML: Patent pending on the use of MELK kinase inhibitors in the treatment of breast cancer, PCT/US2014/010724. AH: Patent pending on the use of MELK kinase inhibitors in the treatment of breast cancer, PCT/US2014/010724, employee of Novartis Pharmaceuticals. JJZ: Patent pending on the use of MELK kinase inhibitors in the treatment of breast cancer, PCT/US2014/010724. JM : Employee of Novartis Pharmaceuticals. LL: Employee of Novartis Pharmaceuticals. FS: Employee of Novartis Pharmaceuticals. RS: Employee of Novartis Pharmaceuticals. The other authors declare that no competing interests exist.

### Funding

| Funder | Grant reference number | Author |
| --- | --- | --- |
| National Institutes of Health | CA172461 | Jean J Zhao |
| National Institutes of Health | P50 CA168504-01A1 | Jean J Zhao |
| National Institutes of Health | P01 CA142536-04 | Jean J Zhao |
| DFCI-Accelerator Fund | | Nathanael S Gray, Jean J Zhao |
| Friends of Dana-Farber Cancer Institute | | Yubao Wang |

The funders had no role in study design, data collection and interpretation, or the decision to submit the work for publication.

### Author contributions

YW, JJZ, Conception and design, Acquisition of data, Analysis and interpretation of data, Drafting or revising the article, Contributed unpublished essential data or reagents; Y-ML, AH, Acquisition of data, Analysis and interpretation of data, Contributed unpublished essential data or reagents; LB, YX, HT, TV, CC, EL, JM, LL, Acquisition of data, Contributed unpublished essential data or reagents; AL, FS, MJE, NSG, Drafting or revising the article, Contributed unpublished essential data or reagents; RS, Analysis and interpretation of data, Contributed unpublished essential data or reagents; TJM, Analysis and interpretation of data, Drafting or revising the article

### Ethics

Animal experimentation: This study was conducted in accordance with the animal use guidelines from the National Institutes of Health and with protocols (#02-127, #06-034) approved by the Dana-Farber Cancer Institute Animal Care and Use Committee.

## Additional files

### Supplementary files

• Supplementary file 1. Breast cancer gene expression datasets used in this study.

• Supplementary file 2. Oligonucleotides used in this study.

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
