## [Decision Letter]

[Editors’ note: this article was originally rejected after discussions between the reviewers, but the authors were invited to resubmit after an appeal against the decision.]

Thank you for choosing to send your work entitled “The oncogenic protein kinase MELK is essential for mitotic progression in basal-like breast cancer cells” for consideration at *eLife*. Your full submission has been evaluated by a Senior editor and 2 peer reviewers, one of whom is a member of our Board of Reviewing Editors, and the decision was reached after discussions between the reviewers. We regret to inform you that your work will not be considered further for publication at this time.

In particular, the reviewers were not convinced that MELK is a bona fide oncogene in basal breast cancer and were further not persuaded that the evidence warrants development of therapies targeting MELK. The functional data provided from a limited series of breast cancer cell lines does not allow generalizable conclusions to be drawn. Moreover, the role of MELK in a wide variety of proliferating cells in many organisms argues against its selective role in basal breast cancer.

Reviewer #1:

The manuscript entitled “The oncogenic protein kinase MELK is essential for mitotic progression in basal-like breast cancer cells” begins with a functional overexpression screen utilizing a library of open reading frames encoding human kinases. The assay was the ability of a partially transformed breast epithelial cell line to form tumors when injected into the mammary fat pad. MELK was one of several kinases that scored in this assay. In several breast cancer gene expression profiling datasets, MELK mRNA expression was higher in the basal subtype, and was associated with poor prognosis and early metastasis – these observations have been made in part by others previously.

MELK protein levels were higher in basal cell lines that in luminal cell lines, although only 2 luminal cell lines were tested. Knockdown of MELK impaired the proliferation of basal cell lines, but had less effect on luminal cell lines, with some caveats (see below). Moreover, MELK knockdown caused tumor regression in xenografts of basal cell lines, but had a more modest effect on xenografts of luminal cell lines. In rescue experiments, kinase-dead MELK was incapable of preventing the toxicity of MELK knockdown. MELK knockdown prevented mitotic progression, leading frequently to apoptosis in basal cell lines. High expression of MELK was correlated with expression of the transcription factor FOXM1. The authors conclude that MELK is a therapeutic target for basal breast cancer, although there is no discussion of the role of MELK in normal cell division and what side effects a MELK inhibitor might be expected to have.

Major comments:

1) The level of interest in these findings relies on whether MELK is selectively utilized for proliferation in basal breast cancer cells versus other proliferating cells. This selectivity is necessary in order to propose that MELK might deserve attention as a target for therapeutic development. The proposed selectivity of MELK for basal breast cancer cells is difficult to reconcile with an essential role for MELK in diverse proliferation systems, including Xenopus oocytes, C. elegans and various proliferating progenitor populations in the mouse (Hebbard et al. Cancer Research 2010 70:8863 and references therein). Indeed, Hebbard et al previously showed that MELK is required for breast cancer tumor initiating cells in a mouse breast cancer model, a fact that is not cited by the authors. Functional studies were restricted to 2 basal and 2 luminal cell lines, so it is difficult to generalize from this small sample set. Moreover, it is unclear whether the knockdown of MELK was equivalent in these experiments between the basal and luminal cell lines since the Western blots were not quantified.

By eye, the degree of knockdown of MELK by shMELK 2 in the basal cell lines (Figure 4) appears stronger than in the luminal cell lines (Figure 4). Getting functional effects by knocking down kinases can be difficult given their enzymatic nature. It could be that the degree of knockdown of MELK in the luminal cell lines did not go below a necessary threshold to cause a proliferation effect. Figure 4 demonstrates that two different MELK shRNAs that have modest differences in their degree of knockdown have very large differences in their effect on proliferation. For these studies to be strengthened, several more cell lines of each type need to be examined and equivalent knockdown needs to be demonstrated by both quantitative protein and mRNA analysis. The xenograft study suffers from the same caveats. In Figure 5—figure supplement 1, the degree of knockdown in the luminal cell lines was not as great as in the one basal cell line presented, which could explain the difference in effect on tumor growth. These xenograft experiments should be repeated under conditions in which MELK knockdown is equivalent, ideally with additional luminal and basal cell lines.

2) The association of MELK with survival in breast cancer is likely due to its correlation with proliferation, which has previously been shown to be a component of gene expression profiles that predicts adverse survival (Dai et al. Cancer Res. 2005 65:4059-66). The authors should determine whether MELK correlates with other published signatures of proliferation and whether its expression adds any statistical power to a proliferation signature in predicting adverse prognosis in breast cancer.

Reviewer #2:

The manuscript describes the dependence on the cell cycle kinase MELK in basal-like breast cancer (BBC) for survival. The authors used a cDNA kinome transformation screen method previously developed in their lab to identify MELK as a potential oncogene. They provided evidence that MELK expression is elevated in BCC and depletion of MELK in BCC cells leads to mitotic defects. Untransformed breast epithelial cells are insensitive to MELK depletion due to low expression levels. These findings indicate that MELK could be a potential target for BCC and other cancer types that show high levels of MELK expression. The data presented in the manuscript is generally of high quality. Previous studies characterizing MELK expression in triple negative breast cancer and other aggressive tumor types somewhat diminish the novelty of this work. The authors showed that the transcription factor FoxM1, which positively regulates MELK expression, is also upregulated in BCC, although the mechanism by which this happens is unclear.

Major points:

1) Informatics for MELK's association with BCC. The authors showed that MELK expression is higher in high-grade breast cancers (Figure 2), thus it is not clear whether MELK expression is higher in in BCC because there are more high-grade tumors in the BCC sub-types than the other 4 sub-types in the dataset used for the analysis. The authors can perform an ANOVA analysis of the TCGA dataset to address this issue. Although the authors suggest that high MELK expression might be a unique phenomenon for BCC, MELK over-expression has been associated with tumor aggressiveness and poor prognosis in several cancer types, including melanoma (Ryu et al. PMID 17611626), glioblastoma and astrocytoma (Nakano et al. PMID 17722061; Marie et al. PMID 17960622), cervical cancer (Rajkumar et al. PMID 21338529), and prostate cancer (Kuner et al. PMID 22945237).

In these prior studies higher MELK expression in more aggressive tumor sample/cell lines is often correlated with other cell cycle genes. Thus it is likely that MELK expression is generally associated with more poorly differentiated tumors that show higher proliferation rate. The authors should discuss their finding in light of these works in other cancer types. Recently MELK has been reported to be over-expressed in TNBC tissues compared to normal ductal cells (Komatsu et al. PMID 23254957), this somewhat diminish the novelty of the authors' finding.

2) The role of MELK in HMEC transformation. The authors identified MELK's transforming activity in HMEC-DD-NeuT cells and demonstrated that MELK can transform Rat1-DD fibroblasts in a Neu-independent fashion. The authors did not present data to show whether MELK can transform HMEC-DD cells without the NeuT oncogene. Since MELK has not been significantly mutated or amplified in breast cancers, the evidence for MELK as a bona fide oncogene is relatively weak.

3) The dependency on MELK in BCC cell lines. Given MELK is a cell cycle regulated genes, it would be important for the authors to test whether MELK expression levels in luminal and basal cell lines (Figure 4) is correlated with their doubling times. Given there are a large number of well-characterized breast cancer cell lines available, testing additional luminal or ER/PR positive cell lines for their dependency on MELK should further strengthen the authors conclusions. MELK was not identified as a BCC lethal gene in the Neel lab's screen using the TRC shRNA library (Marcotte et al., PMID 22585861).

4) The role of MELK in cell cycle. Prior studies indicate that MELK is a cell cycle gene involved in mitosis. It is expressed in tissue progenitor cells and plays a role in embryogenesis. The authors' data indicate that not all proliferating cells require MELK, as knockdown of MELK in MCF7, T47D and HMECs does not impair proliferation. Have the authors investigated whether MELK knockdown in these insensitive cells affect their cell cycle distribution? Have the authors also looked at whether MELK over-expression in HMEC-DD or HMEC-DD-NeuT cells affect their cell cycle and proliferation rate? In xenopus embryo both the loss and over-expression of MELK causes mitotic defects. This would help address the question whether MELK's transforming activity in HMECs is due to a cell cycle effect or a cell-cycle independent mechanism.

---

## [Author Response]

We appreciate the insightful and constructive comments from reviewers and editors. In response to these critiques, we have extensively revised the manuscript, in most cases with new experimental data, including data with a MELK inhibitor showing selective activity against basal-like breast cancer (BBC), the finding of gene amplification of MELK in addition to extensive overexpression of MELK in BBC from a recent TCGA dataset, and a MELK knockout mouse model showing that MELK is dispensable for general proliferation and development. In so doing, we feel that the fundamental conclusions of our manuscript have been significantly strengthened.

*1) Reviewer #1: “The level of interest in these findings relies on whether MELK is selectively utilized for proliferation in basal breast cancer cells versus other proliferating cells*. *This selectivity is necessary in order to propose that MELK might deserve attention as a target for therapeutic development...”*

*Reviewer #2 expresses a similar concern*.

We thank the reviewers for their insightful comments. Indeed, the selective requirement of MELK in basal breast cancer cells versus other proliferating cells is one line of evidence supporting MELK as a therapeutic target. We now have new data from newly developed mouse models demonstrating that MELK is not essential for normal physiological functions.

We have recently generated mice with germ-line knockout (KO) of Melk. Surprisingly, mice with Melk germ-line KO are viable and appear normal without any noticeable phenotypes in the development of the embryos or adult mice. Both male and female mice are fertile and produce litters of normal size. While we were disappointed by these “negative” results, these data suggest that Melk is not an essential gene in mice. This stands in contrast to other mitotic kinases, such as Aurora A (worm ortholog AIR1), Aurora B (worm ortholog AIR2), and PLK1, all of which are essential for the embryonic development of both mice and *C. elegans* (Chase et al., 2000; Schumacher et al., 1998; Severson et al., 2000).

Since bone marrow toxicity is a major side effect of most anti-cancer therapies and, in particular, of those drugs targeting mitotic kinases (e.g., aurora kinases and PLK1), we characterized the immune system in Melk KO mice. We isolated cells from bone marrow, spleen and thymus, and analyzed immune cell populations including monocytes, neutrophils, B and T cells. There were no differences found between wild-type and KO mice (Figure 7). We further analyzed the hematopoietic stem cells and various progenitor cells in the bone marrow of both wild-type and KO mice, and again found no differences between the two groups of mice (Figure 7).

We understand the reviewers’ concern that it is difficult to reconcile the previously observed roles for MELK in diverse proliferating systems, including *Xenopus* oocytes, *C. elegans* and various proliferating progenitor populations in the mouse, with our findings. However, previous studies did not point uniformly to a MELK requirement. For example, while Xenopus MELK (xMELK) is required for the development of *Xenopus* oocytes (30), the *C. elegans* MELK ortholog, PIG-1, is not an essential gene in the worm. Similar to mice, worms with mutated or deleted PIG-1 are viable and relatively healthy albeit with an altered number of certain neurons (7).

A study by [19], where GFP expression was driven by the *Melk* promoter in a mouse model, found a high expression of Melk in mouse mammary progenitor cells, but the functional role of Melk in these cells was not determined. Given these data, we anticipated an impairment of mammary gland development in Melk KO mice. But mammary glands from mice with Melk KO appear completely normal in both morphology and function (e.g., lactation).

Furthermore, the growth of normal human mammary epithelial cells was largely unaffected by shRNA knockdown of MELK with ∼90% efficiency (Figure 5—figure supplement 2).

In summary, while MELK is expressed in a variety of proliferating cells in many organisms, the roles of MELK in these cells and organisms are divergent. Our mouse genetic data strongly suggest that MELK is not essential in mice and deserves attention as a target for therapeutic development. We now have included these data in the revised manuscript.

*2) The functional data provided from a limited series of breast cancer cell lines does not allow broad conclusions to be drawn*.

*Both reviewers consider that since the functional studies were only tested in 2 basal and 2 luminal breast cancer cell lines, the sample set was too small, and testing additional cell lines for their dependency on MELK should further strengthen the authors conclusions. Reviewer #1 also had concerns that the degree of knockdown of MELK by shMELK in the basal cell lines appears stronger than in the luminal cell lines, equivalent knockdown needs to be demonstrated by both quantitative protein and mRNA analysis*.

We agree with the reviewers that quantification of MELK knockdown is important for the evaluation of functional dependence of MELK in these cells. We have re-constructed these luminal cell lines expressing shMELK and were able to achieve an equivalent efficiency of MELK knockdown in these cells (Figure 5—figure supplement 1). Similar to what we observed previously, proliferation of luminal cells were largely unaffected by MELK knockdown (Figure 5).

We also agree that the sample set used in the study was small, and we have expanded it to include 6 basal and 5 luminal breast cancer cell lines. In addition to cancer cell lines, we have also examined MELK expression in primary tumor specimens resected from breast cancer patients. Notably, while robust MELK protein expression was observed in all ER/PR-negative breast tumors, MELK protein was not detectable in these ER/PR-positive breast tumors (Figure 2), suggesting that MELK is largely dispensable for the growth of these luminal tumors.

To complement the RNAi-mediated knockdown approach, we are also using pharmacological inhibition of MELK to evaluate the functional dependency of MELK in basal and luminal breast cancer cells. We tested OTSSP167, a newly developed potent MELK inhibitor (6), on a panel of 4 basal and 4 luminal breast cancer cell lines, and found that all these basal cancer cells are significantly more sensitive to the drug than are the luminal cell lines. OTSSP167 at 100 nM induced dramatic cell death of basal cells, but had little effect on luminal cells (Figure 6—figure supplement 2). Notably, in mice bearing breast cancer cell line-derived tumors, administration of OTSSP167 significantly impaired the growth of basal breast tumors, or even caused tumor regression; by contrast, the growth of luminal tumors were largely unaffected by the treatment (Figure 7). Together, these data indicate that chemical inhibition of MELK preferentially targets basal breast cancer cells. We now have included these new data in the revised manuscript as indicated above.

*3) Since MELK has not been significantly mutated or amplified in breast cancers, the evidence for MELK as a bona fide oncogene is relatively weak*.

*Reviewer #2: “The authors identified MELK's transforming activity in HMEC-DD-NeuT cells and demonstrated that MELK can transform Rat1- DD fibroblasts in a Neu-independent fashion. The authors did not present data to show whether MELK can transform HMEC-DD cells without the NeuT oncogene. Since MELK has not been significantly mutated or amplified in breast cancers, the evidence for MELK as a bona fide oncogene is relatively weak*.*”*

We thank the reviewer for these comments. Regarding the transformation of HMECs, we and others have shown that mutant PIK3CA, Ras or NeuT alone can induce colony formation of HMEC-DD cells in soft agar, but is not sufficient to promote tumor formation of these cells in mice (17; 70; 71). It usually takes two oncogenic events, e.g. PIK3CA plus NeuT or Ras plus NeuT, to fully transform HMEC-DD cells to form tumors in mice. Similar to these potent oncogenes, over-expression of MELK alone can also robustly promote anchorage-independent growth of HMEC-DD cells (Figure 4—figure supplement 2), and again similar to other known oncogenes wild-type MELK cooperates with a second oncogene, e.g., NeuT, to induce these cells to form tumors in vivo.

In contrast to the transformation of HMEC-DD cells, one oncogenic event is sufficient to transform Rat1-DD cells, indicated by both anchorage-independent growth in vitro and tumor formation in vivo (41) . We have shown in our manuscript that MELK overexpression alone in Rat1-DD cells was able to effectively promote colony formation in soft agar or tumor growth in nude mice (Figure 2).

In addition to HMEC-DD and Rat1-DD cells, we also show that overexpression of MELK is able to induced anchorage-independent growth of MCF10A cells (Figure 4—figure supplement 2).

In addition to HMEC-DD and Rat1-DD cells, we also show that overexpression of MELK is able to induced anchorage-independent growth of MCF10A cells (Figure 4—figure supplement 2).

We agree with the reviewer that, in general, genes with hot spot mutations or amplification in human tumors are more readily accepted as oncogenes. Thus, a gene that is over-expressed in tumors needs rigorous testing in relevant systems to demonstrate a causal role in oncogenic transformation. Therefore, we have carried out extensive functional validation of MELK in our study as described above and in the manuscript. In fact, it is striking that the transforming activity of overexpressed MELK, is comparable or superior to that of the highly potent oncogenic PIK3CA-H1047R, in all the transformation assays that we have performed.

With data recently available for the analysis of copy number in large cohort of breast cancer (TCGA, 2012), we find that gene amplification of MELK occurs in breast cancer, especially in ER-negative breast cancer (Figure 3—figure supplement 1). The single ER-positive tumor with MELK amplified is also positive for HER2 expression. Similarly, out of a total 56 established breast cancer cell lines, three are found harboring MELK amplification and are all ER-negative (Figure 3—figure supplement 1). Notably, tumors or cells with MELK amplified tend to express high level of MELK, suggesting that gene amplification contributes to the overexpression of MELK in breast cancer.

Additional points from Reviewer #1:

*The association of MELK with survival in breast cancer is likely due to its correlation with proliferation, which has previously been shown to be a component of gene expression profiles that predicts adverse survival (Dai et al. Cancer Res. 2005 65:4059-66). The authors should determine whether MELK correlates with other published signatures of proliferation and whether its expression adds any statistical power to a proliferation signature in predicting adverse prognosis in breast cancer*.

We agreed with the reviewer that the prognostic feature of MELK expression is likely due to its correlation with cell proliferation. In fact, MELK and other proliferation-related genes are major components of multi-gene signature in predicting disease outcome. For example, a recent study developed a cell proliferation signature composing of top 1% genes whose expression is most positively correlated with proliferating cell nuclear antigen (PCNA) expression, and found that adjusting breast cancer expression data for this cell proliferation signature causes a dramatic reduction in outcome association of most published breast cancer signatures (61). Notably, MELK expression is strongly correlated with cell proliferation, e.g., it’s correlation with PCNA expression in the above-mentioned cell proliferation signature is 10^th^ of the total 129 genes (61).

Additional points from Reviewer #2:

*Informatics for MELK's association with BCC. The authors showed that MELK expression is higher in high-grade breast cancers (*Figure 2*), thus it is not clear whether MELK expression is higher in in BCC because there are more high-grade tumors in the BCC sub-types than the other 4 sub-types in the dataset used for the analysis. The authors can perform an ANOVA analysis of the TCGA dataset to address this issue. Although the authors suggest that high MELK expression might be a unique phenomenon for BCC, MELK over-expression has been associated with tumor agressiveness and poor prognosis in several cancer types, including melanoma (Ryu et al. PMID 17611626), glioblastoma and astrocytoma (Nakano et al. PMID 17722061; Marie et al. PMID 17960622), cervical cancer (Rajkumar et al. PMID 21338529), and prostate cancer (Kuner et al. PMID 22945237). In these prior studies higher MELK expression in more aggressive tumor sample/cell lines is often correlated with other cell cycle genes. Thus it is likely that MELK expression is generally associated with more poorly differentiated tumors that show higher proliferation rate. The authors should discuss their finding in light of these works in other cancer types*.

Following the reviewer’s suggestion, we performed analyses for the correlation of MELK expression with subtypes of breast cancer with the same pathological grade. Through analysis of a large dataset of breast cancer for grades 1, 2 and 3 across all subtypes, respectively, we found that MELK is most highly expressed in BBC (Figure 2—figure supplement 2), suggesting that MELK expression in most pronounced in basal-like breast tumors concerning the same grade of the disease. Moreover, a significant association of MELK expression with disease status also exists within the subtype of BBC (Figure 2—figure supplement 2), suggesting that MELK expression is associated with tumor aggressiveness and poor prognosis in this disease as the reviewer mentioned.

*Recently MELK has been reported to be over-expressed in TNBC tissues compared to normal ductal cells (Komatsu et al. PMID 23254957), this somewhat diminish the novelty of the authors' finding*.

This study, which is based on gene expression profiling of 30 samples of triple negative breast cancer samples and 13 samples of normal breast ductal cells, found that MELK is one of the 301 genes whose expression is upregulated in TNBC. But MELK was not further functionally analyzed, as MELK was not a focus of this study. Our bioinformatics analysis, based on multiple independent large-size cohorts shows that MELK is strongly overexpessed in BBC/TNBC, not only compared to normal breast tissues, but also compared to luminal or ER/PR+ breast cancer. Our data suggest that MELK represent a bona-fide TNBC gene, rather than a general breast cancer gene. More importantly, we have carried out extensive functional analysis of MELK in breast cancer both in vitro and in vivo.

*The dependency on MELK in BCC cell lines. Given MELK is a cell cycle regulated genes, it would be important for the authors to test whether MELK expression levels in luminal and basal cell lines (*Figure 4*) is correlated with their doubling time*.

We thank the reviewer for the insightful suggestion of testing any correlation between MELK expression and the doubling time of cell lines. A recent study from Dr. Joe Gray’s group has systemically analyzed doubling time among 49 breast cancer cell lines. Indeed, basal breast cancer cells have a statistically shorter doubling time (Figure S3A, Heiser et al., 2012). However, we would like to point out that untransformed cells such as HMEC and MCF-10A have much lower expression of levels of MELK than basal breast cancer cells, but have a very similar doubling times. Therefore, it is difficult to conclude on a universal correlation between MELK expression and the rate of cell proliferation.

*MELK was not identified as a BCC lethal gene in the Neel lab's screen using the TRC shRNA library (Marcotte et al., PMID 22585861)*.

As pointed by the reviewer, a recent study (Marcotte et al., 2012) carried out a genome-wide shRNA screen in a total 72 cancer cell lines, including 28 breast cancer cell lines. The criteria applied might explain why MELK was not identified as a breast cancer essential gene. The top-ranked genes from the screens were selected and overlapped against two sets of genes that are likely enriched in essential genes (housekeeping genes (n=1,722), and genes with highly conserved orthologs in eight species (n=1,617)). Interestingly, MELK is not such a housekeeping genes (defined here as expressing in more than 73 of 79 tissues in a human expression compendium), or highly conserved (e.g., MELK ortholog has not been identified in *Saccharomyces cerevisiae*).

*The role of MELK in cell cycle. Prior studies indicate that MELK is a cell cycle gene involved in mitosis. It is expressed in tissue progenitor cells and plays a role in embryogenesis. The authors' data indicate that not all proliferating cells require MELK, as knockdown of MELK in MCF7, T47D and HMECs does not impair proliferation. Have the authors investigated whether MELK knockdown in these insensitive cells affect their cell cycle distribution? Have the authors also looked at whether MELK over-expression in HMEC-DD or HMEC-DD-NeuT cells affect their cell cycle and proliferation rate? In* Xenopus *embryo both the loss and over-expression of MELK causes mitotic defects. This would help address the question whether MELK's transforming activity in HMECs is due to a cell cycle effect or a cell-cycle independent mechanism.*

We thank the reviewer for the insightful comments. We now have performed experiments and found that neither MELK knockdown, nor MELK overexpression in untransformed HMECs alters cell cycle or proliferation as assayed in regular tissue culture. However, the information acquired from regular two-dimensional culturing might not be directly applied to the process of cell transformation, which is typically assayed in three-dimensional conditions. Indeed, when we assessed these cells in suspension culture, overexpression of MELK significantly increased the proliferation of HMEC cells (Figure 4—figure supplement 1). Likewise, MELK overexpression induces colony formation in soft agar (Figure 4—figure supplement 1). Further study is needed to address the correlation of the transforming activity of MELK overexpression with cell cycle.

In summary, recent comprehensive characterization of basal-like breast cancer demonstrates that this subtype of disease has high genetic heterogeneity, but lacks commonly occurring genetic alterations, with the exception of the frequent inactivation of p53 ([Bibr bib5a]). In contrast, the relative uniform overexpression of MELK in basal-like breast cancer makes this gene a unique target if its functional importance can be established. Thus our studies on MELK are critical for guiding the development of targeted therapies in basal-like breast cancer.

Our new study finds that MELK is not an essential gene for many types of normal murine and human cells, but is important for the growth of basal-like breast cancer cells. We plan to add to the current manuscript our data from knockout mice, to strengthen the point that MELK is not a universally essential gene for cell proliferation and only selectively required in basal-like breast cancer. We will also add new data from small molecule studies that use a chemical inhibitor of MELK, to demonstrate the selectivity of targeting MELK in basal versus luminal breast tumors.

The finding of a non-essential mitotic kinase that can be targeted in basal-like breast cancer is novel and unexpected. We are very enthusiastic about this finding, and believe that our work will have enormous impact on guiding the development of therapy for basal-like breast cancer, the most aggressive subtype of breast cancer lacking any proven targeted therapy. In addition, as mentioned by the reviewers, MELK over-expression has been associated with tumor aggressiveness and poor prognosis in several other cancer types, including melanoma, glioblastoma and prostate cancer, though functional studies are lacking. Thus MELK represents a highly promising therapeutic target for BBC and potentially other aggressive tumors.